



# Cirrus cloud shape detection by tomographic extinction retrievals from infrared limb emission sounder measurements

Jörn Ungermann[1,2], Irene Bartolome[1], Sabine Grießbach[3], Reinhold Spang[1], Christian Rolf[1], Martina Krämer[1], Michael Höpfner[4], and Martin Riese[1]

[1]IEK-7, Forschungszentrum Jülich GmbH, Germany
[2]JARA, Forschungszentrum Jülich GmbH, Germany
[3]Jülich Supercomputing Centre, Forschungszentrum Jülich GmbH, Germany
[4]IMK, Karlsruher Institut für Technologie, Germany

**Correspondence:** Jörn Ungermann (j.ungermann@fz-juelich.de)

**Abstract.** An improved cloud index-based method for the detection of clouds in limb sounder data is presented that exploits the spatial overlap of measurements to more precisely detect the location of (optically thin) clouds. A second method based on a tomographic extinction retrieval is also presented. Using CALIPSO data and a generic advanced infrared limb imaging instrument as example for a synthetic study, the new cloud index method is better in detecting the horizontal cloud extent in

comparison to the traditional cloud index and has a reduction of false positive cloud detection events by about 30 %. The results for the extinction retrieval show even an improvement of 60 %. In a second step, the extinction retrieval is applied to real 3-D measurements of the air-borne limb sounder GLORIA taken during the Wave-driven ISentropic Exchange (WISE) campaign to retrieve small-scale cirrus clouds with high spatial accuracy.

# 1 Introduction

Clouds, and in particular cirrus clouds, play an important part in the radiative balance of the atmosphere. The effect of cirrus clouds in a changing climate is still uncertain even though a change in frequency of occurrence is well established due to a redistribution of water vapour in the troposphere (IPCC, 2007; Heymsfield et al., 2017).

To increase our understanding of cirrus clouds, proper observations are required on their frequency, occurrence, coverage,

particle sizes, number concentration, ice water content, and altitude. To generate a sufficient statistical basis for modelling and validation, global measurements of cirrus clouds are required, which can only be generated by satellite-borne instruments (e.g. Krämer et al., 2020). Owing to the dryness of the upper troposphere, cirrus clouds generate often only weak signatures in observations by remote sensing instruments, especially nadir viewing ones. Our knowledge about clouds has been advanced greatly in recent times by the active lidar Cloud-Aerosol Lidar with Orthogonal Polarization (CALIOP) on Cloud-Aerosol

Lidar and Infrared Pathfinder Satellite Observation (CALIPSO; Winker et al., 2009). Due to its active nature, it is the most



highly-resolving and precise satellite instrument with a cirrus cloud product that has been used successfully to create first climatologies (Nazaryan et al., 2008). Other, passive, nadir viewing instruments have often difficulty detecting ultra-thin cirrus or properly determining the top altitude of the clouds with the necessary accuracy to determine, e.g., the radiative effects. While some advances have been made recently, e.g., by Kox et al. (2014), using data from the SEVIRI instrument on MSG (Meteosat Second Generation; Schmetz et al., 2002), these data products currently lack proper error estimates due to the nature of the used classification algorithms. The most sensitive method to detect cirrus from space are in any case provided by limb observing instruments. As cirrus clouds are typically horizontally more elongated than vertically, a limb-viewing instrument has a longer path within the cloud generating a stronger signal and also a much higher vertical resolution. This is true for both occultation and passive emission measuring instruments. While occultation instruments are even more sensitive, passive instruments have the advantage of a much higher measurement density necessary to generate a profound global statistical basis. A historical disadvantage of limb sounders was a bad horizontal resolution compared to nadir viewing ones, but recent developments of tomographic evaluation schemes in combination with proposed instruments with a higher measurement density level the playing field (Griessbach et al., 2020b).

While tomographic retrievals have become state of the art for limb sounders in general (e.g. Steck et al., 2005; Livesey et al., 2006; Christensen et al., 2015), in-orbit instruments do not oversample extensively; e.g., the MIPAS instrument on ENVISAT (Fischer et al., 2008) was designed to have non-overlapping line-of-sights. Hence, this study evaluates the potential capabilities of near-future limb sounders employing imaging detectors with a much higher measurement density, simply called IRLS (infrared limb imaging sounder) in this study. The hypothetical instrument is largely based on the PREMIER IRLS instrument (Process Exploration through Measurement of infrared and millimetre-wave Emitted Radiation; Ungermann et al., 2010; ESA, 2012) proposed to ESA in the Earth Explorer program.

The cloud index is a well proven method for the detection of clouds in infrared spectra (Spang et al., 2001a, 2008, 2012). We show how the spatial resolution and accuracy of the cloud index method can be improved upon by the so called over-sampling, where limb sounders measure spectra so frequently that the line-of-sights of succeeding measurements overlap within the tangent layer. In addition to an enhancement of the cloud index method, we also investigate the capabilities of a tomographic extinction retrieval that employs the same techniques also used for the tomographic retrieval of temperature and trace gases. Previous studies have shown that tomographic methods can increase the horizontal resolution of data products gained from limb sounders up to ideally the spacing between consecutive measurements (von Clarmann et al., 2009; Ungermann et al., 2011; Krisch et al., 2018). Here, we want to investigate how that result typically valid for optically thin conditions transfers itself to optically thicker clouds. To complement the work with synthetic measurements, the final part of this paper applies the extinction tomography to evaluate a tomographic measurement of cirrus clouds made by the GLORIA limb sounder (Riese et al., 2014; Friedl-Vallon et al., 2014). This air-borne instrument points in a 90° angle in relation to the heading of its carrier and has thus a very different measurement geometry compared to along-track-pointing satellite instruments. In this sense it is a very hard test case for the method as this geometry makes the retrieval much more involved and complicated.





This paper is structured as follows. We will first present briefly the employed data products and the models and instruments they were derived from. Section 3 presents the new algorithms and methods that are then studied in depth in Sect. 4. We conclude with a first 3-D tomographic cloud retrieval based on real GLORIA measurements in Sect. 5.

## 2 Instruments and Data

### 2.1 MIPAS

The Michelson Interferometer for Passive Atmospheric Sounding (MIPAS; Fischer et al., 2008) was an infrared Fourier-transform spectrometer aboard the ESA satellite Envisat. It measured the spectral range from $685\,\mathrm{cm}^{-1}$ to $2410\,\mathrm{cm}^{-1}$ with a spectral resolution of up to $0.025\,\mathrm{cm}^{-1}$. We employ here the reduced spectral resolution of $0.0625\,\mathrm{cm}^{-1}$ that was used in its last operational years. The vertical sampling was $\approx 1.5\,\mathrm{km}$ in the UTLS. We assume for the synthetic simulation of MIPAS measurements a horizontal sampling of $420\,\mathrm{km}$ that was used in the period from 2005–2012 and the vertical field-of-view, which has a full-width-half-max of roughly $0.06°$ (equivalent to about $3\,\mathrm{km}$ vertically; e.g., von Clarmann et al., 2003).

### 2.2 GLORIA and IRLS

The Gimballed Limb Observer for Radiance Imaging in the Atmosphere (GLORIA; Riese et al., 2014; Friedl-Vallon et al., 2014) is an airborne imaging Fourier-transform spectrometer, capable of acquiring more than 6000 interferograms with its $128 \times 48$ used detector pixels in less than 2 seconds. This enables it to measure a full atmospheric profile at once. The horizontal dimension across the line of sight is currently only used to increase the signal-to-noise ratio by averaging, but could be exploited as well, e.g., for imaging small-scale cloud structures. The interferogram acquisition time can be adjusted between fast measurements with coarse spectral resolution and slow measurements with a high spectral resolution over the effective spectral range from $780\,\mathrm{cm}^{-1}$ to $1400\,\mathrm{cm}^{-1}$ (Kleinert et al., 2014). The spectral sampling is configurable between $0.625\,\mathrm{cm}^{-1}$ (roughly $2\,\mathrm{s}$ acquisition time) and $0.0625\,\mathrm{cm}^{-1}$ (roughly $10\,\mathrm{s}$ acquisition time). The effective spectral resolution is roughly a factor of two worse than the sampling due to the employed Norton-Beer apodization (Norton and Beer, 1976). A unique feature of GLORIA is its capability to point its detector towards different directions compared to the aircraft heading. This allows for the measurement of airmasses between $45°$ and $\approx 132°$ with respect to flight direction. By constantly panning the instrument over the available angles, the same airmasses are measured from multiple directions, which enables the tomographic reconstruction of three-dimensional structures (Ungermann et al., 2011; Krisch et al., 2017, 2018).

GLORIA has been operated successfully during multiple campaigns on both the German HALO research aircraft and the Russian M-55 Geophysika. The measurements discussed later were taken on the 18th September 2017 during the Wave-driven ISentropic Exchange (WISE) campaign based in Shannon, Ireland.

The Infrared Remote Limb Sounder (IRLS) is an Earth-observing satellite instrument that was originally proposed for the 7th ESA Earth Explorer program. It is, effectively, a GLORIA-like instrument in space with a fixed viewing direction backwards compared to its flight direction. This configuration also allows for tomographic 3-D reconstruction of the measured

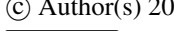



atmosphere. While the PREMIER IRLS offered higher spectral resolutions, we focus here on the spatial capabilities and thus
assume a spectral sampling of $1.25\,\mathrm{cm}^{-1}$ and 15 horizontal measurement tracks covering $7°$ as well as an along-track sampling
of $50\,\mathrm{km}$. For the vertical sampling, we assume a pixel pitch of $0.014°$, which corresponds roughly to a vertical sampling of
$\approx 700\,\mathrm{m}$ in the troposphere. These capabilities are in line with those of the GLORIA instrument and thus certainly achievable.

### 2.3 CALIOP

The Cloud-Aerosol Lidar with Orthogonal Polarization (CALIOP; Winker et al., 2007, 2009) is a nadir viewing lidar on the
CALIPSO satellite, which is part of NASA's A-train. CALIOP provides high resolution vertical profiles of cloud and aerosol
properties with a vertical resolution of $60\,\mathrm{m}$ below $20.2\,\mathrm{km}$ and $180\,\mathrm{m}$ above and an along track resolution of $5\,\mathrm{km}$. We used
cloud extinction data from the L2CPro V3.01 product files (CALIPSO, 2018) of the full month of December 2009 to generate
test cases of realistic 2-D cloud scenes for the limb viewing geometry using a radiative transfer model (see Sec. 3). In addition to
the given extinction values, we also used a data set where we reduced the supplied extinction values by an order of magnitude.
This allows us to explore the sensitivity of limb sounders with respect to clouds that are thinner than those present in CALIOP
level 2 products.

### 2.4 ECMWF

For pressure and temperature in the cloud scene simulations based on CALIOP data and as a priori in all retrievals, ERA-Interim
data provided by the European Centre for Medium-Range Weather Forecast (ECMWF; Dee et al., 2011) was employed. The
model data is available in $6\,\mathrm{h}$ time steps with the T255/L60 resolution, which corresponds to a horizontal sampling of $\approx 80\,\mathrm{km}$.
Quadri-linear interpolation is used for resampling the model onto the needed grids, whereby pressure is interpolated in log-
space. The horizontal wind speeds and diabatic heating rates from the ECMWF ERA-Interim data were also used for the
calculation of backward trajectories using CLaMS.

### 2.5 CLaMS-ICE

As CALIOP only provides vertical 2-D slices, we turned towards simulations generated by the Chemical Lagrangian Model of
the Stratosphere (CLaMS; McKenna et al., 2002; Konopka et al., 2007; Ploeger et al., 2010) for providing realistic 3-D cloud
structures. The CLaMS-ICE module (Luebke et al., 2016) includes a double-moment bulk microphysics scheme for modelling
cirrus clouds (i.e. ice water content and ice crystal number; Spichtinger and Gierens, 2009). The box model runs in forward
direction on backward trajectories (24 hours) started from points on a regular grid with user-defined resolution and extent
in longitude, latitude, and pressure space. A resolution $0.25°$ in the horizontal and $0.5\,\mathrm{km}$ in the vertical domain is used for
resolving finer cirrus structures compared to the original ERA interim resolution. For initialisation, cloud ice water content and
specific humidity from ERA interim are spatially interpolated to the CLaMS-ICE starting point of each individual trajectory.



The trajectories are calculated on hybrid-potential temperature coordinates, which allows to resolve transport processes in
the troposphere influenced by the orography and transport processes in the stratosphere, where adiabatic horizontal transport
dominates.

To reduce the computational effort of the radiative transfer model, we transform the ice water content and radius information
supplied by the CLaMS-ICE module to a simple extinction coefficient by the formula of Gayet et al. (2004): $E = A \cdot I \cdot R^{-1}$,
with $E$ being extinction in $\mathrm{km}^{-1}$ at $804\,\mathrm{nm}$, $A = 1500\,\mathrm{mm}^3\mathrm{g}^{-1}$, $I$ the ice water content in $\mathrm{gm}^{-3}$, and $R$ the radius of particles
in µm.

## 3   Methods

### 3.1   JURASSIC2

This paper employs the JUelich RApid Spectral Simulation Code version 2 (JURASSIC2), which is a radiative transfer code
optimized for large-scale and tomographic simulations and retrievals (Hoffmann, 2006). It includes capabilities from computing
spectrally resolved radiances line-by-line to using spectrally averaged lookup-tables for extremely fast computations. The
algorithmic adjoint model allows its efficient use in retrieval schemes (employing the JUelich Tomographic Inversion Library;
JUTIL) and data assimilation in general. JURASSIC2 has been exemplary used for the analysis of CRISTA-NF data (e.g.
Kalicinsky et al., 2013), for the study of clouds and aerosol using MIPAS data (e.g. Griessbach et al., 2013, 2014), for the
operational processing for the GLORIA instrument (Ungermann et al., 2015, e.g.) and for studies on the aerosol layer in the
Asian Summer Monsoon (Höpfner et al., 2019).

In this study, the emissivity-growth-approximation (e.g. Weinreb and Neuendorffer, 1973; Gordley and Russell, 1981) is
used in combination with pre-calculated lookup tables of optical path (also called optical depth, or thickness) in relation to
temperature, pressure, and volume mixing ratio to quickly compute radiances and derivatives with respect to temperature in a
discrete representation of the atmosphere.

Simulations with JURASSIC2 for MIPAS-like and IRLS spectra employed the trace gasses $CCl_4$, $CFC-11$, $H_2O$, $HNO_3$,
and $O_3$ with climatological values (Remedios et al., 2007). A ray-tracing step length of $5\,\mathrm{km}$ was employed. For both MIPAS-
like and IRLS measurements, the spectral samples from $787.50\,\mathrm{cm}^{-1}$ to $796.25\,\mathrm{cm}^{-1}$ and from $831.25\,\mathrm{km}^{-1}$ to $835.00\,\mathrm{cm}^{-1}$
were used, albeit with different spectral sampling for each instrument of $0.0625\,\mathrm{cm}^{-1}$ and $1.25\,\mathrm{cm}^{-1}$, respectively, and a strong
Norton-Beer apodization. We simulated synthetic MIPAS-like and IRLS radiances for all CALIOP extinctions of December
2009 and these microwindows.

### 3.2   Cloud index

A practical method for identifying a radiance measurement of a cloud is the so called cloud index (CI), first introduced by Spang
et al. (2001b). The CI is defined as the ratio between a spectral region with a strong emission feature such as the $CO_2$ Q-branch
at $12.6\,\mathrm{\mu m}$ and an atmospheric window such as the one located at $12\,\mathrm{\mu m}$. It is a dimensionless quantity with a slight dependence





on latitude and season. For given tangent point altitudes, one can derive specific thresholds that separate cloudy measurements from others. Typical CI values in cloudy conditions are between 1.1 and 4 (Spang et al., 2012, 2015) indicating optically thick to thin conditions. Here, we employ altitude dependent thresholds of Sembhi et al. (2012) to determine if an index indicates a cloud. For optically thin conditions, the CI correlates well with extinction and the integrated volume density or area density path along the limb path (Spang et al., 2012). Latter differentiation depends on the particle radius range, where larger median

radii, typical for ice clouds, correlate with the area density. Although the CI approach is an effective and computational cheap detection mechanism for thin and thick clouds, its information content is limited when retrieving cloud information below the cloud top, as clouds affect the CI of clear air below and hence, the cloud detection thresholds are not effective in distinguishing clear air from cloud below the cloud top.

### 3.3    Cloud extent retrieval

This section describes two different approaches to the spatial detection of ice clouds from measured infrared limb spectra. The first approach builds on the CI method (e.g. Spang et al., 2012), which can detect the presence of a cloud in a measured spectrum. Previous satellite instruments, such as MIPAS-Envisat (Fischer et al., 2008), have a comparatively coarse measurement pattern, where individual lines-of-sight of measured spectra do not meaningfully intersect (at least in the most common operation modes). Figure 1 shows the line-of-sights of measurements of a MIPAS-like instrument and the assumed IRLS. At lower

altitudes, the line-of-sights of MIPAS do not overlap at all in the tangent layer. In contrast, the IRLS has a very fine measurement grid, where many line-of-sights overlap to the extent that the line-of-sights of the immediately neighboring measurements of one altitude overlap at their tangent point altitude. This overlap of the IRLS lines-of-sight allows for the application of tomographic methods (e.g. Carlotti et al., 2001; Livesey et al., 2006) and even requires them to utilize the instrument to its full capacity.

165       The first proposed spatial detection method is a two-dimensional evolution of the CI method, described in Sec. 3.3.1. The cloud detection is improved by taking into account not only the tangent point of a measurement but instead the full extent of and overlap within the tangent point layer. This extension already allows for the exploitation of the increased sampling density of IRLS like instruments.

   In addition, a much more computationally involved method is proposed in Sec. 3.3.2. This method deduces the atmospheric

extinction values of clouds in a tomographic non-linear inversion that has so far been mostly used to derive temperature and trace gas concentrations.

### 3.3.1    2-D convex hull CI

The principal problem of the CI method is that one does not know at which point of the line-of-sight of a measurement a potential cloud is located. Sometimes even a cloud far above the tangent point layer generates a CI value below the threshold.

This section introduces a method that exploits the existing overlap of measurements of limb sounder instruments to improve upon the positioning of detected clouds, i.e. to properly compute the convex hull of clouds determined by the CI of individual spectra. As the CI is computed from integrated micro-windows over a rather large spectral range, there is no meaningful effect





on the results by the spectral resolutions of IRLS measurements. Therefore, this aspect of different instruments is neglected (further, the increased spectral resolution typically comes at the cost of a reduced horizontal sampling, which counteracts the purpose of exploring the spatial detection capabilities).

A common approximation of the spatial origin of a radiance measurement is assigning it to its tangent point, which is the location along its line-of-sight that is closest to the surface. This is the location where the air is densest and thus from where most radiation is emitted in optically thin conditions. Obviously, this assumption breaks down in the presence of clouds. This poses the largest problem in determining the position of clouds from the CI and the tangent point location alone. Figure 2 shows an extinction cross-sections and associated CI values. Comparing the location of small clouds (extinctions $> 10^{-3}\,\mathrm{km}^{-1}$) in the upper panels to the corresponding structures in CI in the lower panels, one can easily see how the assumption of emission stemming from the tangent point breaks down for an optically thick (i.e., intransparent) medium. The curved structures peak at the location of the cloud and then extend downwards to both sides for measurements that either 'hit' the cloud before or after the tangent point. The resulting structure is still useful as it is a strict overestimate of the dimensions of the clouds (above the detection limit). Also the cloud top altitude of the true cloud can be properly determined to high accuracy compared to, e.g., nadir sounders.

Exploiting that measurements indicated as cloud free according to the CI have typically no cloud within the tangent point layer, where the lines-of-sight are nearly horizontal, we can improve the method. Especially thin ice clouds are often rather thin vertically, which means that the radiances measured by lines-of-sights passing through the thin cloud at steeper angles may not be strongly affected and thus not detect its presence. Therefore, one may not easily extend the cloud-free assumption to the layers that the line-of-sight passes through at altitudes significantly above the tangent altitude.

The proposed first new detection method works as follows:

1. Build a regular grid covering the cross-section. Here, a grid with a vertical spacing of 500 m was chosen to be a bit finer than the measurement grid of the IRLS to be robust against slight variations of tangent point altitude due to temperature and pressure variations. The horizontal location of the profiles was taken to coincide with the lowermost tangent points (this causes a slight shift between grid and tangent point location for higher altitudes). Each grid box is assigned a value of 0 (i.e. is assumed to be cloudy; final values of zero can also be used, however, to determine grid boxes without measurement information).

2. The line-of-sight for each spectrum is computed for a distance of $d\,\mathrm{km}$ before and behind its tangent point. Here, a very conservative (small) value of 100 km was chosen for $d$ to reduce the number of false-negatives (i.e., to decrease the number of undetected clouds).

3. Successively for each line-of-sight, each grid box that it passes through is assigned the maximum between its current value and the CI of the spectrum.

4. Last, the CI of each grid box is compared to the CI threshold associated with its altitude and latitude band (Sembhi et al., 2012) to determine whether or not a cloud is present.





This algorithm gives a cloud/no-cloud decision for the 2-D cross-section measured by the limb sounder that is more precise than the CI method alone. For one cross-section of a half-orbit as shown below (e.g., Fig. 4), the method uses less than 1 minute of computation time on one core of a current system.

### 3.3.2 Tomographic extinction retrieval

The second method investigates briefly into the capabilities of employing a full-blown non-linear retrieval for the determination of cloud positions. This is a computationally more demanding task compared to the color-ratio based schemes and as such may be less suited for a quick identification scheme for filtering affected spectra. However, computational capacity steadily increases and the current scheme is well suitable for real-time usage.

The method is comparable to the one employed by Castelli et al. (2011), but simplified due to the neglecting of scattering.
The intent is to show the capabilities of this approach in combination with an increased measurement density. The retrieval employed the same JURASSIC2 forward model and setup that was used for generating the synthetic measurements. While the simulated measurements were generated using the original, fine grid on which the CALIOP L2 data are supplied, the retrieval grid was reduced to $500\,\mathrm{m}$ vertically in the relevant altitude range and $\approx 20\,\mathrm{km}$ horizontally. This corresponds to roughly 1000 profiles for the half-orbit in the CALIOP based simulations. We use the same spectral setup as used for the generation
of synthetic radiances, i.e. only two averaged radiances were simulated centered at $796.25\,\mathrm{cm}^{-1}$ and $835.00\,\mathrm{cm}^{-1}$. The same trace gases and volume mixing ratios were used in the retrieval as in the forward simulation. Perfect knowledge was assumed for all trace gases, obviously a strong simplification. But due to the strong radiative effect even of thin ice clouds in the limb, this is likely justified, but beyond the scope of this study to examine. Temperature and extinctions were assumed unknown and climatological values and a zero profile were used as a priori and initial guess for temperature and extinction, respectively.
Also, the scattering effect of clouds was neglected here, as we are interested only in the detection, not in a quantitative analysis of the retrieved extinction.

Computing the spatially located extinction from the radiances poses an inverse problem. The solution to this problem is identified by iteratively modifying an atmospheric state $\boldsymbol{x}_i$ such that the simulated measurements $F(\boldsymbol{x}_i)$ progressively agree better with the actual (in this case also partially simulated) measurements $\boldsymbol{y}$ within expectation to the noise equivalent spectral
radiances (NESR) of the measurements under the side-condition of being close to a 'plausible' atmospheric state $\boldsymbol{x}_\mathrm{a}$:

$$
\begin{aligned}
\boldsymbol{x}_{i+1} \;=\; \boldsymbol{x}_i &- \left(\mathbf{S}_\mathrm{a}^{-1} + \mathbf{F}'(\boldsymbol{x}_i)^\mathrm{T}\mathbf{S}_\epsilon^{-1}\mathbf{F}'(\boldsymbol{x}_i) + \lambda_i\mathbf{I}_n\right)^{-1} \cdot \\
&\left(\mathbf{S}_\mathrm{a}^{-1}(\boldsymbol{x}_i - \boldsymbol{x}_\mathrm{a}) + \mathbf{F}'(\boldsymbol{x}_i)^\mathrm{T}\mathbf{S}_\epsilon^{-1}\left(\boldsymbol{F}(\boldsymbol{x}_i) - \boldsymbol{y}\right)\right).
\end{aligned}
\tag{1}
$$

The matrix $\mathbf{S}_\epsilon$ is set up assuming a 0.1 % error in radiance. The matrix $\mathbf{S}_\mathrm{a}^{-1}$ is defined as a Tikhonov-Phillips type regularization matrix imposing smoothness conditions in horizontal and vertical direction on the solution (Tikhonov and Arsenin, 1977).
For temperature, only the second derivative is constrained with correlation lengths of vertically $1\,\mathrm{km}$ and horizontally $200\,\mathrm{km}$. Restraining the second derivative enforces a smooth lapse rate for temperature (Ungermann et al., 2015), which is useful for further analysis of the dynamical structure around the thermal tropopause. For extinction, the first derivative is constrained in



addition to a weak regularization towards the zero profile. These values are selected to be similar to those practically used in
tomographic studies for the GLORIA instrument. The qualitative result does not depend largely on the type of regularization
as long as it is neither too strong to smoothen the solution nor too weak to allow for oscillations, as the aim is so far not to
perfectly reproduce the original extinction values, but to arrive at a simple cloud/no-cloud product. The retrieval employs the
numerical techniques developed for the GLORIA limb sounder tomography to quickly derive a solution (Ungermann et al.,
2010, 2011). As only 2 spectrally averaged samples are simulated from each spectrum, the computation time and memory
consumption are manageable. The retrieval for a half-orbit consumes about 200 MiB, mostly for storing Jacobian matrices and
requires about 25 minutes on 8 cores to converge to a satisfactory state, which can be readily accomplished in real-time.

## 4 Study on synthetic data

In this section we use synthetic spectra generated by JURASSIC2 based on CALIOP extinctions to evaluate the algorithms.

### 4.1 2-D convex hull CI

In a first step, we compare the cloud indices as gained from MIPAS-like and IRLS spectra. Figure 2b shows the CI for the
simulated spectra based on an exemplary CALIOP cross-section and the MIPAS measurement grid and spectral resolution.
The simulated radiances were generated using the grid defined by the CALIOP L2 data shown in Fig. 2a. It is obvious that the
'pixels' are rather coarse compared to the fine structure of the clouds contained in the CALIOP data. Figure 2c shows the IRLS
simulations. The increased spatial sampling density results in a much finer sampling of the clouds, but the bow-like artifacts
due to the optically thick atmospheric conditions become apparent. These are also given in the synthetic MIPAS data but barely
discernible due to the coarse measurement grid. Figures 2d-f show the same situation, but with CALIOP extinction reduced by
a factor of ten. These numerical experiments shift the focus to very thin clouds that may not be present in the original data set.

The results of the convex hull CI algorithm are exemplary depicted in Figs 3 and 4. Figure 3 shows the results for the MIPAS
instrument, while Fig. 4 shows the result for the IRLS instrument. For MIPAS, no obvious differences between CI and convex
hull CI are apparent. The minor visible discrepancies can be attributed to the difference between the tangent point grid of
the CI and the rectilinear grid on which the convex hull CI algorithm operates. In contrast, a noticeable improvement can be
seen for the IRLS measurements. The convex hull CI algorithm reduces the bow like structures around thin clouds. Especially
the comparatively small structures at 38°N have a much reduced sized more aligned with the true structure as shown by the
extinction contour. But a deficit is also apparent. Below thick clouds, no actual measurement information is present and the CI
is (wrongly) attributed at the tangent point location. This will still cause an overestimation of cloud presence.

### 4.2 2-D tomographic extinction retrieval

This section presents some results of the extinction retrieval. Figure 5 shows the extinction retrieved for the same extinction
distribution derived from CALIOP data for MIPAS and IRLS data for one of the 440 processed cross sections. Both retrievals
were setup identically with the difference that Fig. 5b used simulated MIPAS measurements and a coarser horizontal resolu-



tion ($\approx 160\,\mathrm{km}$) and Fig. 5c used simulated IRLS measurements. The MIPAS-based retrieval is both horizontally and vertically

coarser in comparison to the IRLS retrieval as required by the different measurement grid. The results are generally more accurate than the picture provided by the CI in Fig. 3 as even for the coarse MIPAS measurement grid, the overlap of measurements at higher altitudes can be used to constrain the location of thin clouds better. The retrieval using the IRLS measurement specification in Fig. 5c offers a much better resolved result. The higher spatial sampling and the reduced FOV allow us to see below clouds with clear sky conditions as with the equatorial high cirrus cloud. We encountered similar conditions also with

our air-borne instruments GLORIA (see below) and CRISTA-NF (Spang et al., 2008). The location of clouds, expressed in increased values of extinction, is more precise and much closer to the actual extinction distribution compared to the CI based methods. Some artifacts remain below thick clouds, where, due to their opacity, no or next to no measurement information is present. As such the values below 9 km at 20°S are not reliable. On the other hand, the weak structures at and above 15 km are well reproduced. Obviously, the retrieval works better for optically thin conditions.

### 4.3 Comparison of 2-D cloud top detection accuracy

This section aims to quantify the performance of the 2-D convex hull CI and the 2-D tomographic extinction retrieval algorithms with respect to cloud top height estimation. We focus here on the capabilities of the IRLS instrument. Figure 6 shows an exemplary CALIOP orbit with detected cloud extent according to the CALIOP extinctions (extinction larger than $10^{-4}\,\mathrm{km^{-1}}$), cloud index (CI), convex hull cloud index (convex hull CI) and tomographic extinction retrieval (extinction

larger than $3 \cdot 10^{-4}\,\mathrm{km^{-1}}$). A slightly larger threshold is employed here, as the smoothing by the regularization extends the rather large extinctions vertically and a smaller threshold would thus lead to a systematic overestimation of cloud extent. One can immediately see that all methods typically agree within about 1 km, with larger errors occurring at the border of extended cloud regions.

Table 1 shows the numerical results for the cloud top height derived by the three algorithms for 440 semi-randomly selected

CALIOP orbits (we arbitrarily picked one month of data). Using unmodified CALIOP extinctions, the CI shows a positive bias of 1.08 km with a standard deviation of 2.29 km. There are two major sources for a high bias. First, the field-of-view of the instrument causes an overestimation of cloud top altitude, especially for thicker clouds (e.g. Griessbach et al., 2020a). Second, and more importantly, the cloud is horizontally extended causing cloud detection events beside the actual cloud, where no cloud is in the original data (e.g. Kent et al., 1997). The second effect mainly causes the large variance in the results. As expected,

the convex hull CI algorithm significantly reduces this bias, whereby the tomographic extinction retrieval seems to be even superior. Both are capable of reducing the impact of "horizontal cloud lengthening". For thinner clouds, the situation improves all around. Using clouds that are an order of magnitude thinner, here, the CI shows a bias of only 0.66 km. For optically thinner clouds the general cloud top height overestimation is reduced or even turns into an underestimation (Griessbach et al., 2020a) and the "horizontal cloud lengthening" is also reduced (Fig. 5). The bias of the convex hull CI algorithm and the tomographic

extinction retrieval are both in the order of 160 m, whereby the extinction retrieval has a reduced standard deviation compared to the convex hull algorithm.





As the overestimation of the horizontal extent of clouds strongly affects the cloud top altitude comparison, we examined more closely how well the shape of the cloud top is reproduced. In a first step the true cloud top is determined from CALIOP extinctions. Using the tomographic extinction retrieval grid, all cloud top grid boxes plus all boxes within two squares distance

(Manhattan norm, ±1 km vertically, ±50 km horizontally) are selected for comparison. The results are collected in Tab. 2. The table shows an increase for correctly identified cloudy pixels for the three methods with the conventional CI being worst and the tomographic extinction retrieval being best. False positives decrease accordingly. However, there is a slight increase for false negatives, which is caused by horizontally small clouds that are filtered away by the convex hull CI as the CI is pushed below the threshold and for which the tomographic extinction retrieval determines an extinction below the threshold (potentially due

to slightly smearing out the cloud). This is asserted by the numbers for the scaled extinction test cases, where the amount of false negatives increases over the board

### 4.4    3-D IRLS scenario — Spatial detection capabilities

This section describes the result for the convex hull algorithm CI for the CLaMS-ICE model based simulations that allow the simulation of the across-track coverage of the IRLS instrument in contrast to the CALIOP based simulations that allow only a

simulation of the center track.

Figure 7 shows horizontal cross-sections at several pressure levels through the extinctions derived from the 3-D CLaMS-ICE ice water content simulation. The center of the simulated IRLS measurements follows the 19°W meridian. Actual satellite measurements of the instrument will not follow a perfect polar orbit, but for the simulations, the difference is negligible and simplifies the simulation setup. One can see a cloud field on the left-hand-side on lower altitudes and a second cloud field on

the right-hand-side at different altitudes. At higher latitudes, the two cloud fields merge. This situation gives a nice across-track variation over the images of the IRLS that can be examined in the following.

This variation can be better seen in Fig. 8 that shows the CLaMS-ICE extinction as 'images' as the IRLS would see it. Each 'pixel' of the picture corresponds to one pixel of the IRLS, but the horizontal coverage is slightly larger by 2 pixels. The IRLS would take roughly twice as many images of the situation than depicted here. The depicted images cover the latitudes

from 39.1°N to 64.6°N. The simulation ends shortly beyond these latitudes. The images show a two-layered structure on the left-hand-side between 42.7°N and 51.9°N. Northward of 46.4°N one can see the second cloud structure to the right with first rather faint extinctions and then higher ones until the two structure join around 61°N.

The measurements of the individual tracks can be treated individually as singular cross-sections and may be assembled in a second step. As MIPAS only measured a single track, no MIPAS simulations are shown here for comparison as the results will

not be different from those of Sect. 4.1.

The computation of the conventional CI and the application of the cloud index threshold is depicted in Fig. 9a and b. Similar to the simulations of Sect. 4.1, the cloud extent is overestimated to the sides of the clouds and below. The result of the convex hull CI algorithm is shown in Fig. 9c and d. One can see again, that the new algorithm follows the true extinction closer. In the case of the central track (Fig. 9e), it becomes apparent that the chosen value for extinction of $10^{-3}\,\mathrm{km}^{-1}$ is not the true

detection limit as the faint cloud structure in the center track below that limit is also detected by the CI.





The individual tracks can then be assembled into a three-dimensional view on the cloud structure. Figure 10a shows a three-dimensional representation of the true extinction distribution. The zonal extent is limited by the measurement coverage of the IRLS. One can see the across-track and along-track variation as well as the vertical structure. The result of the convex hull CI algorithm is presented in Fig. 10b. The three-dimensional results agree similarly as the cross-sections discussed before. The

cloud extent is slightly overestimated horizontally and the vertical structure of the cloud is lost, respectively, the bottom of the cloud is often located lower than in the actual extinction structure. Also, the very small spot of a cloud to the south was not detected.

## 5   3-D retrievals using GLORIA measurements

The final section applies the extinction retrieval approach on real measurements. While currently no limb sounding satellite

instrument with a sufficient measurement density in the UTLS exists, the airborne GLORIA instrument can serve well for a feasibility study, even though the 3-D retrieval of GLORIA is more complicated than the one needed for satellite-borne instruments. In fact, it resembles more closely techniques to reconstruct 3-D cloud structures from ground based cloud imagers (Mejia et al., 2018), but with a horizontal viewing geometry and a moving single instrument instead of multiple stationary ones.

This numerical experiment uses measurements acquired on the 18th September 2017 during the WISE campaign. Here, the GLORIA instrument operated with a spectral sampling of $0.2\,\mathrm{cm}^{-1}$ and panning from $45°$ to $132°$ in $6°$ steps, while following a straight flight path.

We used all measurements taken between 11:10 UTC and 12:35 UTC in the reconstruction. All spectral samples between $832.4\,\mathrm{cm}^{-1}$ and $834.4\,\mathrm{cm}^{-1}$ were averaged and used as single micro-window for the extinction retrieval. We used only a single

window here, as we assumed that ECMWF supplies temperature with a sufficient quality such that we do not need to retrieve it. We further drastically simplify the retrieval by not taking trace gas emissions into account. Measurements with tangent points below $\approx 9\,\mathrm{km}$ altitude were discarded as they do not contribute to the reconstruction of the cirrus clouds at higher altitudes. Thus, 629 separate images with in total 61 611 radiance values were employed in the retrieval. Figure 11 shows one exemplary cloud scene. While the camera in the visible samples a viewing angle of about $11°$ and shows an extended cirrus cloud at

$\approx 10.5\,\mathrm{km}$, the infrared camera samples only a section of about $1.5°$. Here, the employed horizontal averaging over the IR pixels is useful, but other imaged clouds exhibit finer structures within an image, similar to the filament at $319.3°$ azimuth. But, even with the fine retrieval grid described below, they would remain inaccessible. Future work would encompass a measurement scheme that reduces the horizontal gaps between images and exploits the full resolution capabilities of the detector.

The retrieval grid used a vertical sampling of $125\,\mathrm{m}$ and a horizontal sampling of $10\,\mathrm{km}$ in both horizontal directions. The

grid is rectilinear in a stereographic projection centered around the center point of the volume rotated in such a fashion that one axis of the grid is parallel to the flight path. The grid covered the volume of $\pm 1\,000\,\mathrm{km}$ in the horizontal direction and between $8\,\mathrm{km}$ and $16\,\mathrm{km}$ altitude in the vertical direction. The grid was extended further to encompass the whole measured volume





up to 64 km altitude at a reduced sampling[1]. Altogether, this resulted in 3 235 925 extinction values to be reconstructed. This number is significantly larger than the number of measurements, making this a drastically under-determined problem.

To compensate for the under-determination, we employed regularization of zero-th and first order. We used a standard deviation for extinction of $10^{-3}\,\mathrm{km}^{-1}$ for atmospheric samples with an ECMWF potential vorticity below 3 PVU and $5 \cdot 10^{-5}\,\mathrm{km}^{-1}$ for atmospheric samples with an ECMWF potential vorticity above 5 PVU. In between a linearly interpolated value was used. This setup avoids strong cloud signals in the stratosphere, which might otherwise appear as artifacts outside the well-measured volume and it doesn't affect the reconstructed extinctions in the well-resolved region. We assumed extinction

structures to be typically ≈100 times longer horizontally than vertically and scaled the first derivative accordingly. The well-resolved volume surrounds the locations of the tangent points and is highlighted in the retrieval results. See Krisch et al. (2018) for a more involved discussion on this kind of linear-flight tomography and its capabilities.

Figure 12 shows the measured brightness temperatures taken at different azimuth angles. Just the three given angles give already some insight into the real cloud structure. The thin structure at 10.5 km at 11:45 UTC in Fig. 12a is the increased

radiance emitted by a cirrus cloud. From this figure alone, it is not determined if the cloud extends vertically over several kilometres or moves away from the aircraft at earlier measurements. The images taken at different azimuth angles deliver the missing information. The image taken at 126° shows the cloud shrunken to nearly a single blob. This tells us the angle of the elongated cirrus cloud with respect to the flight path. Taken together, we can compute that this cloud runs nearly in a perfect North-South direction. In combination with the slanted structure of the cloud at other angles the horizontal extent orthogonal to

the flight path can be computed. Further information could be discerned from the vertical structure at 11:45 UTC in Fig. 12b. A similar structure is given in all measurements taken at this time. This is caused by a cloud very close to the aircraft as it is seen at all azimuth angles around this same time, but not present at times more than a couple of minutes before and after. The retrieval is a mathematical method to extract this and more information from the measurements in an optimal fashion.

The volume can best be reconstructed close to the tangent points of the radiance measurements (e.g. Krisch et al., 2018).

Atmospheric samples much closer to the flight path were not measured at all. Atmospheric samples beyond the volume covered by tangent points could also be reconstructed, but with quickly deteriorating quality. As in all tomographic reconstructions, measured structures are smeared along the line-of-sights of the measurements if not constrained by measurements taken at different angles. Due to the curvature of the Earth, thick clouds measured at low altitudes are smeared along the line-of-sight, which curves upwards from the tangent points and causes high extinction values at implausible altitudes — which is one of the

reasons we employed the PV-dependent regularization scheme. Please note that due to the geometry of satellite measurements, the overlap of the line-of-sights is much better, which prevents such artefacts. The results of the reconstruction are depicted in Fig. 13. The two horizontal cross-sections show retrieved extinction values at two altitudes. At 10.5 km, one can see two cloud structures close to Iceland, a vertically thick and horizontally extended cloud at 20°W and the thin cirrus cloud previously discussed at 16°W. The North-South extension already visible from Fig. 12 is also present here. Neither of the clouds visible

at 10.5 km extends up to 12.5 km, where several small clouds close to the flight path are reproduced by the retrieval, which are associated to the vertically elongated areas of increased radiance in Fig. 12.

---

[1] vertically, 2 km steps until 24 km, 8 km steps until 64 km; horizontally, 1000 km steps until 3 000 km distance from the center





A theoretical analysis of the achieved resolution gives similar results to our previous work on deriving temperature structures (Krisch et al., 2018). Some systematic uncertainty due to uncertainty in line-of-sight might bias the cloud tops estimate[2]. The vertical resolution in the area of the cirrus clouds is in the order of $300\,\mathrm{m}$ decreasing to $400\,\mathrm{m}$ for lower levels. The horizontal

resolution is in the order of $30\,\mathrm{km}$ in flight-track-direction and $70\,\mathrm{km}$ orthogonal to that. Circular flight patterns, or better, a backwards-viewing limb satellite could further improve the horizontal resolution (Ungermann et al., 2011; Krisch et al., 2017).

While the vertical extent, especially the cloud top, can be deduced with very high precision compared to nadir viewing instruments, it is not obvious that the horizontal extent was properly derived. For verification and comparison of quality of both, we used nadir viewing images of the Spinning Enhanced Visible and Infrared Imager (SEVIRI; Schmetz et al., 2002) on

the second generation operational weather satellite Meteosat (MSG) and the Moderate Resolution Imaging Spectroradiometer (MODIS; Platnick et al., 2015, Platnick et al., 2017) on the Terra satellite. Due to its geostationary orbit, SEVIRI offers a good temporal coverage. Two spectral channels are shown in Fig. 14. The image from the visible spectrum at $0.8\,\mathrm{\mu m}$ shows scattered light from clouds at all altitudes. The second channel at $12.0\,\mathrm{\mu m}$ shows the brightness temperature of infrared light emitted by the ground and clouds (while scattering also plays a role here, we believe its influence to be small enough such that

we can neglect it for this qualitative discussion). The 12 micron channel was selected as it uses the same spectral region that is also used in the GLORIA extinction retrieval. Clouds at higher altitude have a smaller brightness temperature than clouds at lower altitudes due to their lower temperature. Both images were taken at 12:00 UTC, which corresponds roughly to the mid-point of GLORIA measurements. In addition, we used the cloud top product from the MODIS instrument that passed over our measurement area in low Earth orbit around 13:08 UTC, which is only slightly beyond GLORIA's measurement period,

and still reasonably close to compare the cloud top altitudes, if not necessarily horizontal position that might have been shifted due to advection. Figure 15 shows the level 2 cloud top altitude product.

We focus first on region A in Figs 13, 14, and 15. The GLORIA radiance and extinction data shows here an optically thick cloud within the jet-stream vertically ranging over several kilometers topping out at $\approx 11\,\mathrm{km}$. This agrees with the MODIS cloud top data (mostly 10.5km with some pixels slightly above 11.0km) and is also consistent with the low brightness temper-

atures ($<260\,\mathrm{K}$) in the SEVIRI 12 micron band. In the horizontal, both, SEVIRI and MODIS see the high altitude cloud in the same position filling approximately the same area in region A and indicating no significant horizontal movement between both measurements. However, the GLORIA measurements fill a larger fraction of region A, which we attribute to the horizontal and temporal smearing of the retrieval (also here, a fast moving satellite would be subject to less temporal smearing due to a changing scene). The SEVIRI time series shows that the front is moving quickly eastwards and the 12:15 UTC image agrees

already much better with our data (please not that the images containing most information were taken around 12:22 UTC pointing at $126°$ in relation to aircraft heading).

Second, we compare the structures found in region B in the same figures. There is a cloud stretching in north-south direction for several $100\,\mathrm{km}$ with a brightness temperature of $272\pm1\,\mathrm{K}$ according to SEVIRI. The structure and location of the cloud visible in SEVIRI data compares favourably with a cloud located at the same horizontal position at 10 to $10.75\,\mathrm{km}$ altitude in

---

[2]current estimates of our pointing accuracy are in the order of 0.05-0.1°, which translates to a bias of $50\,\mathrm{m}$ $5\,\mathrm{km}$ below the aircraft, getting progressively worse towards lower altitudes.





the GLORIA extinction retrieval (only one layer is depicted in Fig. 13). The magnitude of derived extinction indicates that the
cloud is quite transparent in nadir view such that the brightness temperature of SEVIRI is certainly largely caused by warmer
air and haze of lower altitudes. This cloud ($\approx 1\,\mathrm{km}$ thick with an extinction of $\approx 0.1\,\mathrm{km}$) should reduce the measured nadir
brightness temperature by $\approx 4\,\mathrm{K}$, which is consistent with the difference of $3\,\mathrm{K}$ observed by SEVIRI between the cloud and
surrounding air. Please note that the cirrus cloud is well visible in the optical regime in Fig. 11. Also MODIS sees a quite similar

cloud, but assigns it a cloud top altitude of below $1000\,\mathrm{m}$ consistent with the high brightness temperature visible in SEVIRI.
For these thin layers of cirrus, a limb sounder provides much higher accuracy in cloud top determination than state-of-the-art
cloud top products derived from nadir sounders (Weisz et al., 2007).

    Third, region C is discussed. In contrast to the other, larger clouds, GLORIA detected very small clouds bringing it to its
spatial detection limits as these clouds are small enough to fall into the gaps of the horizontal scans. However, the measurements

indicate very thin cirrus clouds at an altitude of 12 to $13\,\mathrm{km}$. The retrieval assembles the measurements to a region with small,
spotty clouds of differing optical thickness with a top altitude of $12.75\,\mathrm{km}$, which coincides with the cold point tropopause
having here a temperature of $\approx 210\,\mathrm{K}$. While SEVIRI and MODIS show similar small patchy clouds in region C, it is more
difficult to assign the high altitude clouds retrieved from GLORIA. The cloud feature at the southern tip of region C agrees
with a cloud feature measured by SEVIRI and MODIS. While the low brightness temperature of SEVIRIs $12\,\mu\mathrm{m}$ channel

indicates a high altitude cloud, as in region A, the MODIS cloud top altitude is below $2\,\mathrm{km}$. From this discrepancy we deduce
that these patchy clouds are probably too small and/or optically thin for IR nadir measurements to properly assign a cloud
top altitude. While some of these have a comparably low brightness temperature of $265\,\mathrm{K}$ (the bright spot at the lowermost
corner of region C), it is quite different from the $210\,\mathrm{K}$ corresponding to the altitude of the clouds detected by GLORIA. Due
to the location close to the flight path, it could even be that the clouds that are visible in SEVIRI are unrelated clouds at lower

altitudes as they would be outside the field-of-view of GLORIA. While the large cloud of region B was detected by MODIS
albeit at a much too low altitude, these even thinner clouds have likely been missed totally. There are no high clouds in the
MODIS data in region C and it is not easy to construct a relationship between the very small low clouds in region C in MODIS
and our high clouds.

## 6    Conclusions

We presented a new cloud index product, the convex hull cloud index, which can exploit the higher measurement density of
current and future over-sampling limb sounders. While it cannot improve upon the cloud identification for older limb-sounders
such as MIPAS, it offers a significantly better cloud identification for higher measurement densities. We could show that it
can properly locate clouds along the line-of-sight for many cases involving cirrus-clouds and thus reduce the number of false-
positive cloud detection events by about $30\,\%$. This method does not require a radiative transfer model and is computationally

very cheap.

    In addition, we introduced a tomographic extinction retrieval for cloud detection based on recent advances in retrieval
techniques for limb sounders. The method uses the same algorithms and models used for 3-D temperature and trace gas





reconstructions. In its current state, the required computational time is small compared to the measurement time thus allowing for real-time application. The tomographic extinction retrieval generated a statistically better result compared to the color ratio

methods with a reduction of false positive detection events of more than 60 % compared to the standard cloud index. It excels in optically thin conditions, but could deal well with all typical cirrus clouds in the upper troposphere. For optically thick conditions given at lower altitudes the algorithms are naturally limited by the lack of information in the measurements by the limb sounders. Here, synergy with available nadir sounders could be exploited.

We finally applied the extinction retrieval to real measurements by the GLORIA limb sounder and could reconstruct sev-

eral high and low altitude clouds in three dimensions. The vertical and horizontal position of these clouds was compared to images of the SEVIRI instrument and the cloud top altitude product based on MODIS data. We found good agreement in the general structure of the detected thick cirrus clouds. For thinner cirrus clouds, horizontal extent agreed, but vertical positioning disagreed. For very small and high cirrus clouds no strong correlation was found between the three products. Determining the cloud top altitude of the reconstructed clouds from GLORIA measurements is easily feasible with an accuracy of less than

300 m while the horizontal positioning is less certain. Due to the tomographic measurement principle a resolution of 30 to 70 km can be achieved, which may be further worsened depending on advection due to strong winds.

Summarising, we have shown that tomographic reconstruction schemes applied to densely sampled limb sounding observations allow to extract a wealth of information on high clouds, reaching beyond what standard methods can achieve. It demonstrates that these kind of observations are well suited to collect information on high clouds not being accessible by other

kind of global measurements.

*Data availability.* The simulations and retrievals can be requested from the author.

*Competing interests.* No competing interests are present.

*Acknowledgements.* The authors are grateful to ECMWF for providing operational analysis and forecast as well as reanalysis data through the MARS server. Parts of this research was supported by the Deutsche Forschungsgemeinschaft, DFG, in the *Cirrus clouds in the extra-*

*tropical tropopause and lowermost stratosphere region (CiTroS)* project, project number SP 969/1-1 and by ESA in the *Characterisation of particulates in the upper troposphere / lower stratosphere* project (Contract No: 400011677/16/NL/LvH). The authors gratefully acknowledge the computing time granted through JARA on the supercomputer JURECA Jülich Supercomputing Centre (2018) at Forschungszentrum Jülich. The authors acknowledge also the teams of MODIS and SEVIRIS for providing their data products. The GLORIA measurements and retrievals are based on the efforts of all members of the GLORIA team, including the technology institutes ZEA-1 and ZEA-2 at

Forschungszentrum Jülich and the Institute for Data Processing and Electronics at the Karlsruhe Institute of Technology. We would also like to thank the pilots and ground-support team at the Flight Experiments facility of the Deutsches Zentrum für Luft- und Raumfahrt (DLR-FX).



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





| Cloud top altitude comparison | | | |
|---|---|---|---|
| Testcase | CI error | convex hull CI err. | ext. ret. error |
| CALIOP extinctions | $1.08\pm2.29$ km | $0.71\pm2.03$ km | $0.47\pm1.50$ km |
| $10^{-1}$ CALIOP extinctions | $0.66\pm2.14$ km | $0.16\pm1.96$ km | $0.16\pm1.32$ km |

**Table 1.** This table aggregates the difference between true cloud top altitude and determined cloud top altitude for 440 CALIOP orbits acquired in December 2012 for simulated IRLS measurements.

| Cloud top shape comparison | CI | | | convex hull CI | | | ext. ret. | | |
|---|---|---|---|---|---|---|---|---|---|
| Testcase | ok | fn | fp | ok | fn | fp | ok | fn | fp |
| CALIOP extinctions | 74 | 1 | 24 | 80 | 4 | 16 | 89 | 4 | 7 |
| $10^{-1}$ CALIOP ext. | 78 | 4 | 18 | 81 | 8 | 12 | 89 | 6 | 5 |

**Table 2.** This table aggregates the difference between true cloud top shape and determined cloud top shape for 440 CALIOP orbits acquired in December 2012 for simulated IRLS measurements. All values in percent. 'ok' means correctly detected, 'fn' is a false negative, 'fp' a false positive.



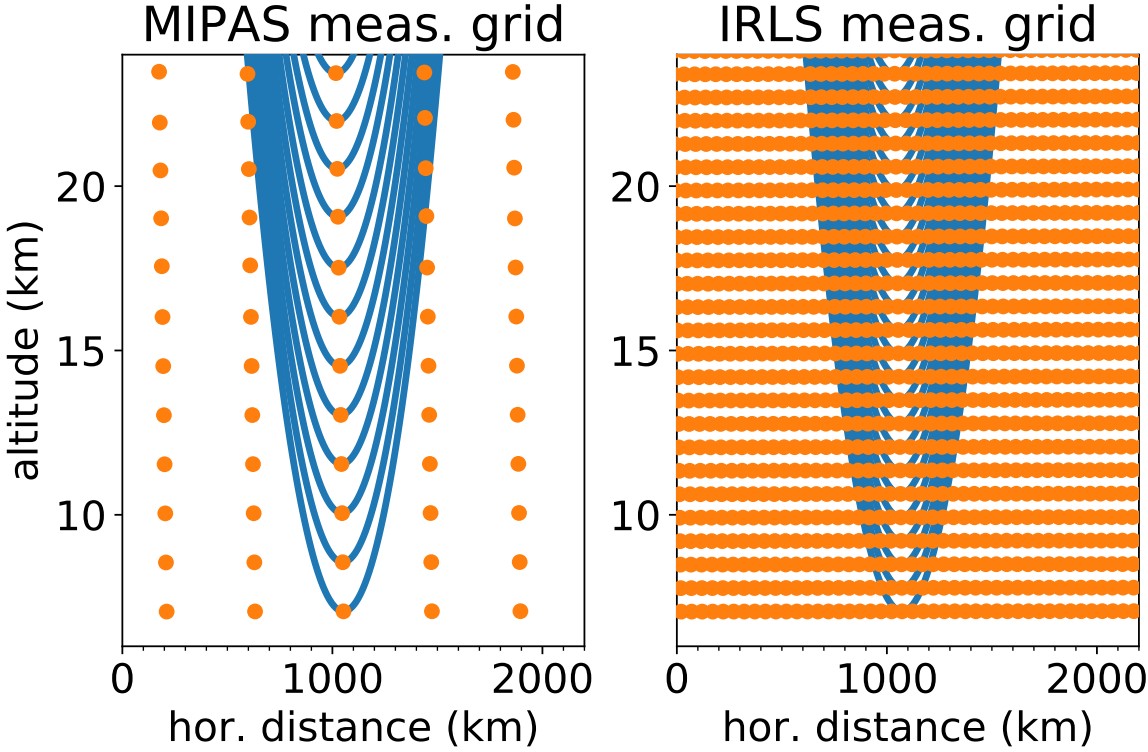

**Figure 1.** The measurement grid of a MIPAS-like instrument (left panel) and the IRLS (right panel). The blue lines indicate the line-of-sights of a single profile. The orange dots indicate the tangent points of multiple adjacent profiles 2100 km along-track.



**Figure 2.** CALIOP extinction **(a)** and CALIOP extinction reduced by a factor of ten **(d)** as well as the associated CI derived from simulated MIPAS measurements in Panels **(b)** and **(e)**. The CI derived from simulated IRLS measurements is given in Panels **(c)** and **(f)**, respectively. The contour line for an extinction value of $10^{-3}\,\mathrm{km}^{-1}$ is shown as a light blue line in all panels.



**Figure 3.** Result of the convex hull CI algorithm for MIPAS simulations based on CALIOP extinctions reduced by a factor of ten. The top panel **(a)** shows the CI at the location of tangent points. Panel **(b)** shows the location of clouds according to the altitude dependent CI threshold. Panel **(c)** shows the CI determined with the convex hull CI algorithm. Panel **(d)** shows the location of clouds according to the altitude dependent CI threshold. The cloud position from Panel (b) is shown as a red contour line for reference. The contour line for a 'true' extinction value of $10^{-3}\,\mathrm{km}^{-1}$ is shown as a light blue line in all panels.



**Figure 4.** Result of the convex hull CI algorithm for IRLS simulations based on CALIOP extinctions reduced by a factor of ten. The top panel **(a)** shows the CI at the location of the tangent points. Panel **(b)** shows the location of clouds according to the altitude dependent CI threshold. Panel **(c)** shows the CI determined with the convex hull CI algorithm. Panel **(d)** shows the location of clouds according to the altitude dependent CI threshold. The cloud position from Panel (b) is shown as a red contour line for reference. The contour line for a 'true' extinction value of $10^{-3}\,\mathrm{km}^{-1}$ is shown as a light blue line in all panels.



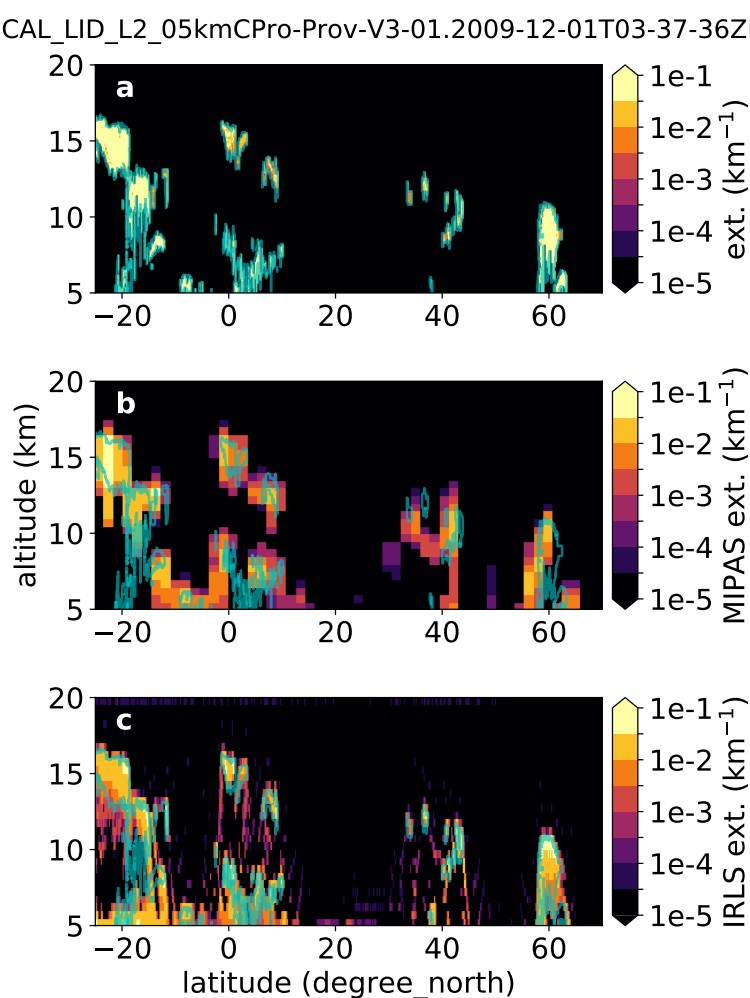

**Figure 5.** Retrieval of extinction from simulated measurements using CALIOP extinctions reduced by a factor of ten. Panel (**a**) shows the true extinction while Panel (**b**) shows the retrieved values for a MIPAS like instrument. Panel (**c**) gives the results for an instrument with higher measurement density such as the IRLS. The contour line for a 'true' extinction value of $10^{-3}\,\mathrm{km}^{-1}$ is shown as a light blue line in all panels.

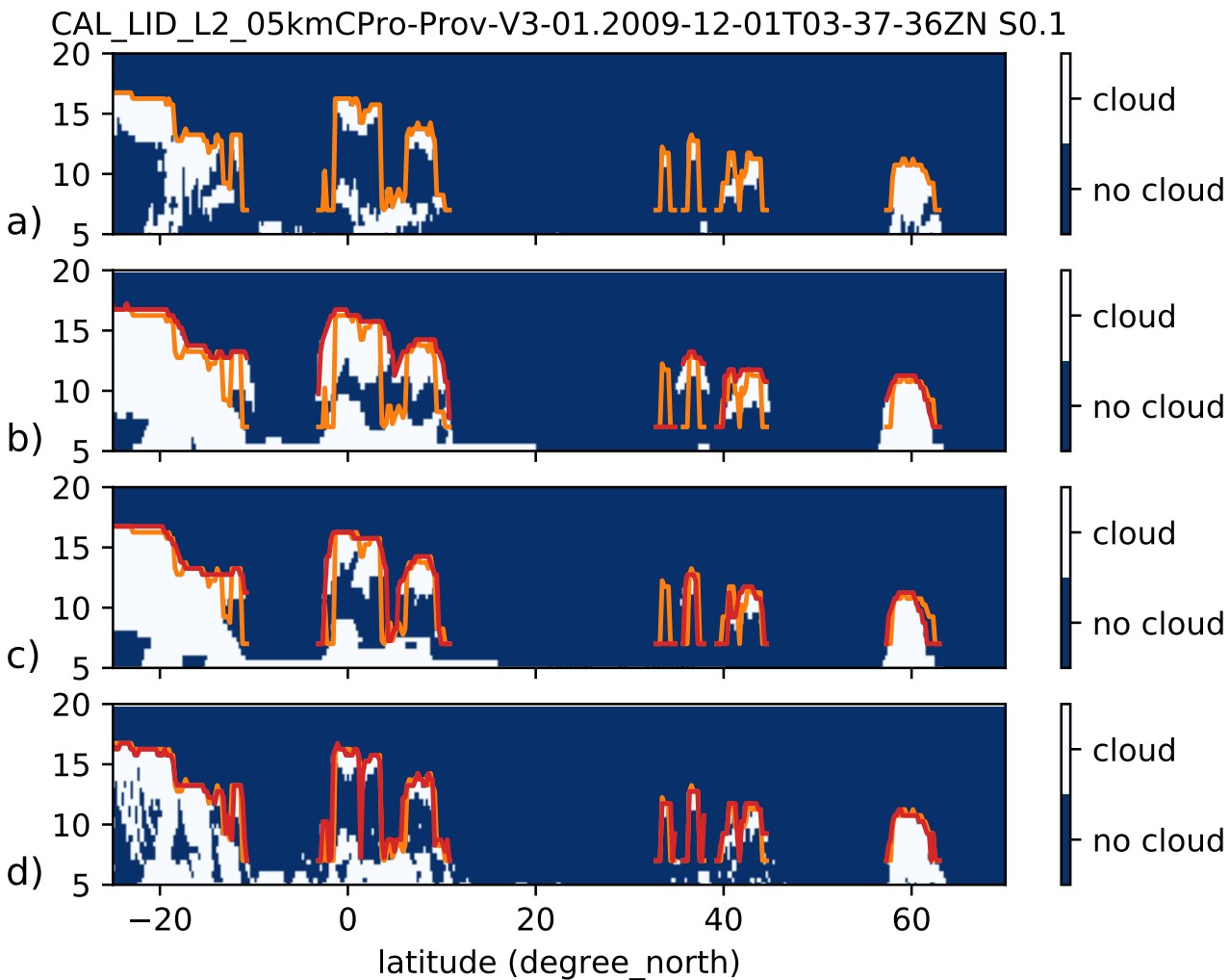

**Figure 6.** Comparison of cloud top detection capability for the various discussed methods and the IRLS instrument. Panel **(a)** shows the cloud extent contained in the CALIOP data. Panel **(b)** shows the cloud extent according to the CI. Panel **(c)** shows the cloud extent according to the convex hull CI algorithm. Panel **(d)** shows the cloud extent according to the tomographic extinction retrieval. In all panels, the orange line shows the true cloud top altitude, where clouds are present above 7 km with a lower limit of 7 km. The red line shows the cloud top altitude according to the depicted algorithm.



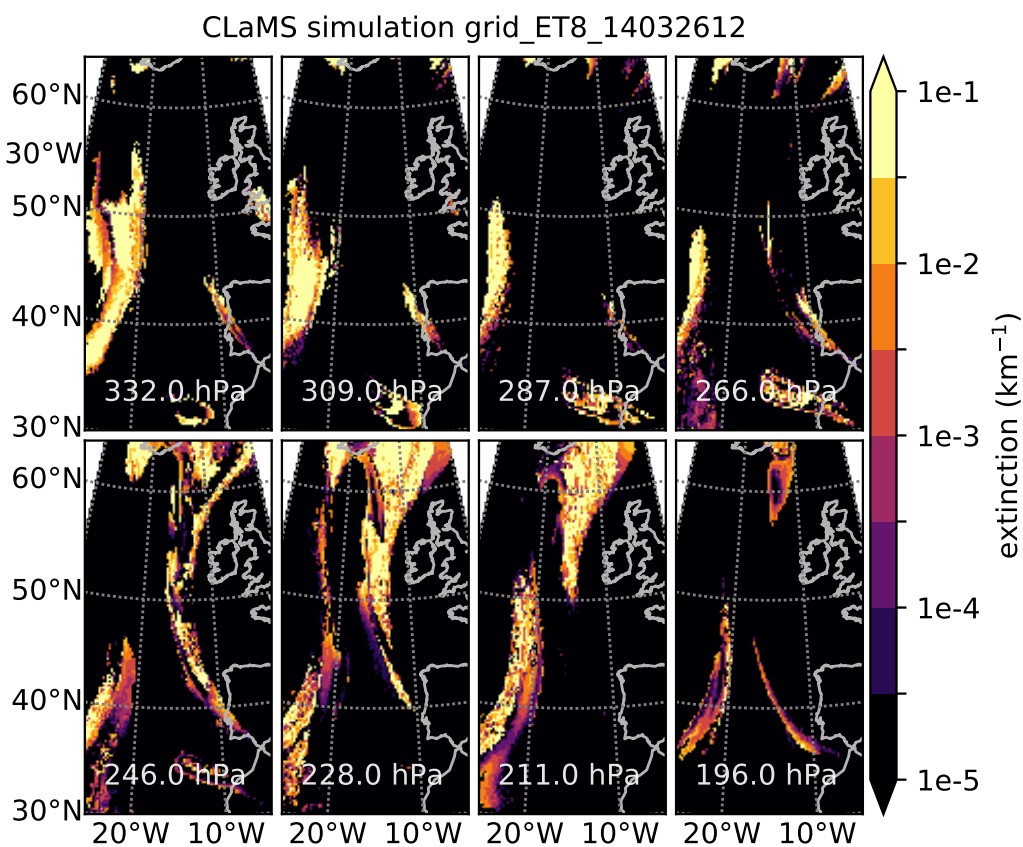

**Figure 7.** Extinction of a cirrus cloud simulated by CLaMS-ICE and computed from ice water content and particle radii.



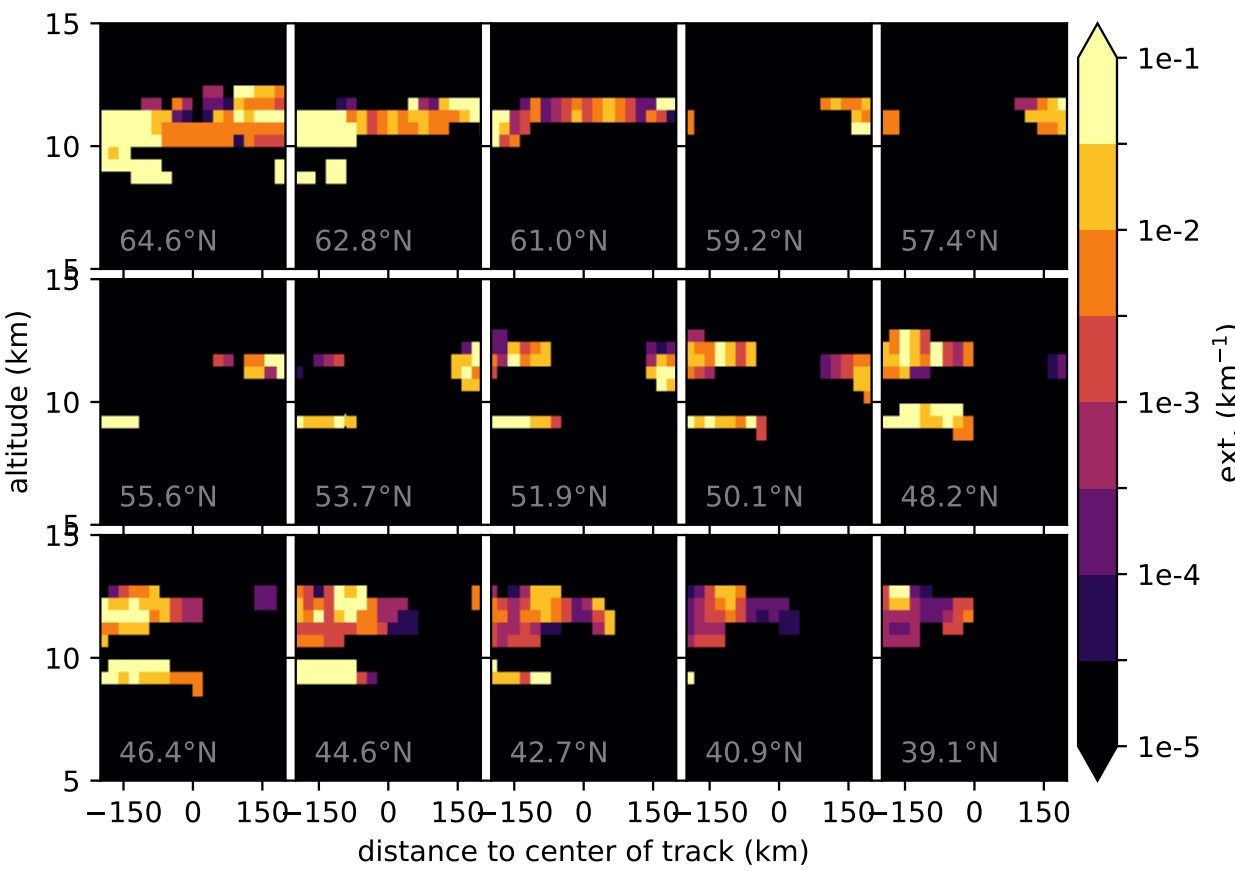

**Figure 8.** Cross-section through CLaMS-ICE extinction data corresponding roughly to the field-of-view of the IRLS. Only every second image of the IRLS is depicted.

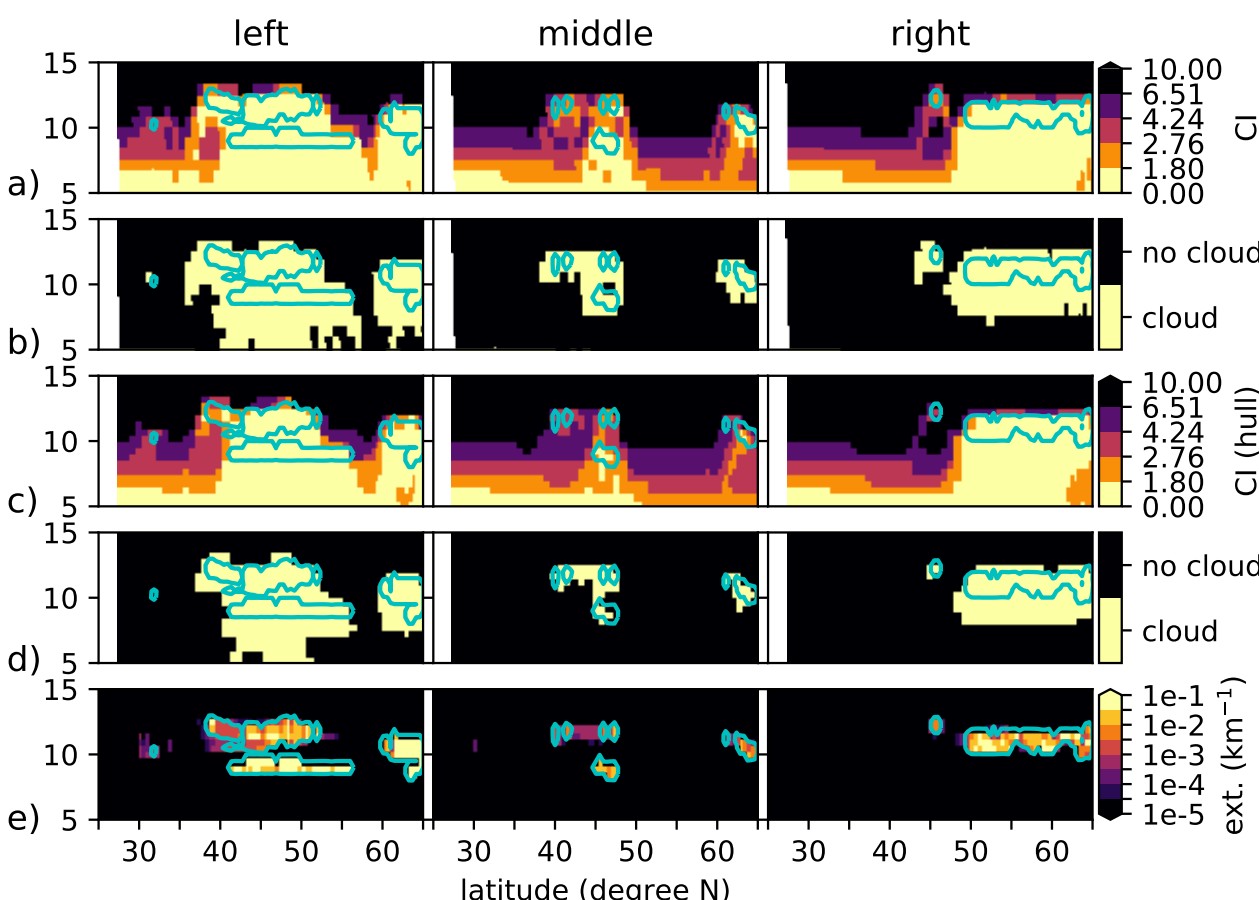

**Figure 9.** Cloud index and convex hull CI for three measurement tracks of a simulate IRLS instrument (left-most, center, and right-most). Panel **(a)** shows the cloud index, Panel **(b)** shows the corresponding cloud detection. Panel **(c)** shows the CI as derived from the convex hull CI algorithm, whereas Panel **(d)** shows the cloud detection for the convex hull CI. Panel **(e)** shows the extinction values used for generating the simulated measurements. The contour line for a 'true' extinction value of $10^{-3}\,\mathrm{km}^{-1}$ is shown as a light blue line in all panels.



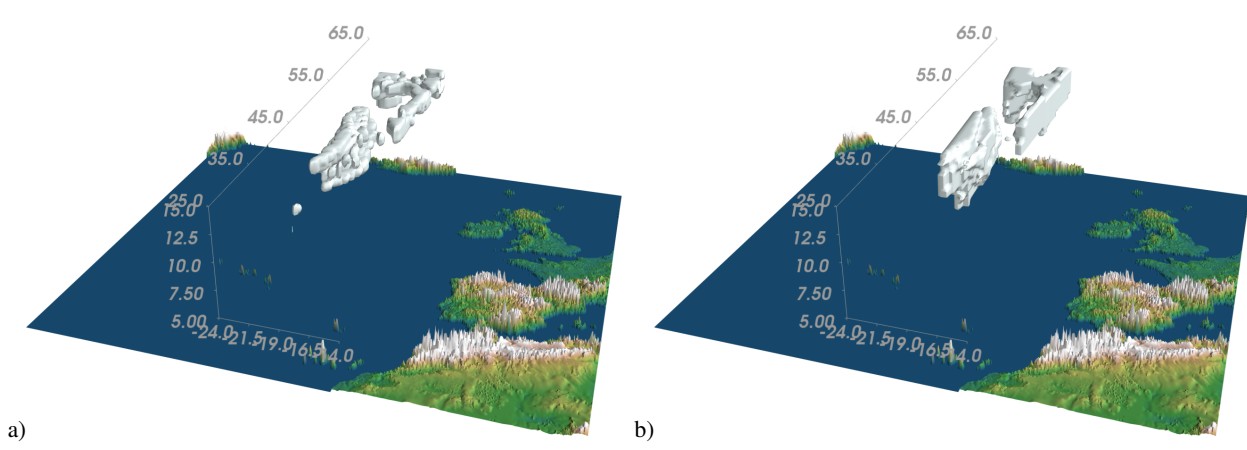

a)                                                    b)

**Figure 10.** The true (panel (**a**)) and derived (panel (**b**)) cloud structure using the convex hull CI algorithm. Shown is a white contour surface for $10^{-3}\,\mathrm{km}^{-1}$ in the measurement track of the IRLS. The vertical dimension is stretched by a factor of roughly 100. Due to the employed cylindrical projection, the measurements expand towards the back.

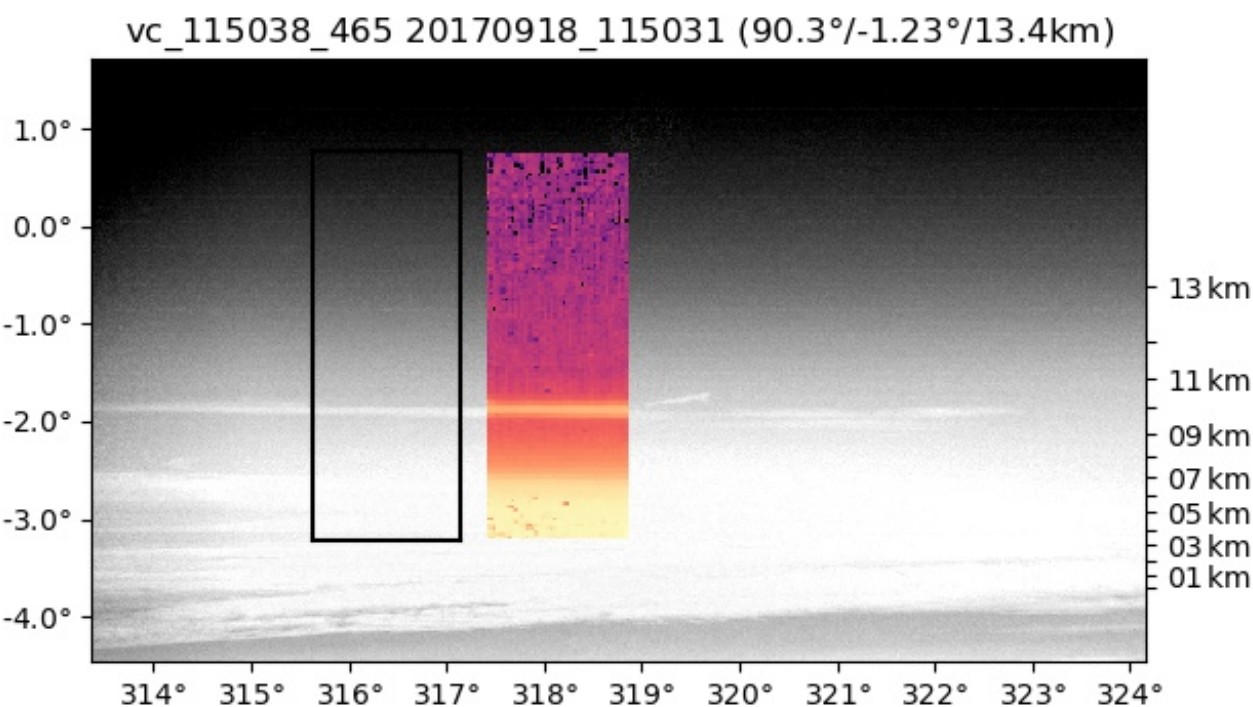

**Figure 11.** A superposition of a visual and infrared image taken at 11:50:31 UTC showing a cirrus cloud located at 10.5 km. The visual image is similar to a normal camera and has a much wider field-of-view than the infrared one. The infrared image has been shifted to the right for better comparability and the radiance is depicted in a logarithmic scale with arbitrary units; the black frame marks the original position. The altitude axis on the right gives an approximate tangent point altitude.





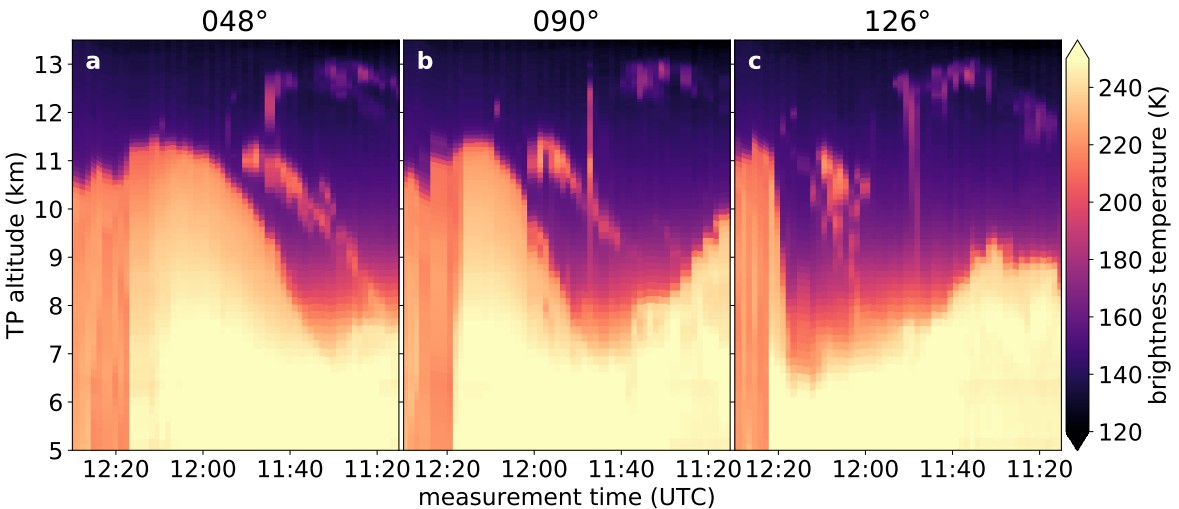

**Figure 12.** Brightness temperature over time sorted according to azimuth angle in relation to aircraft heading.

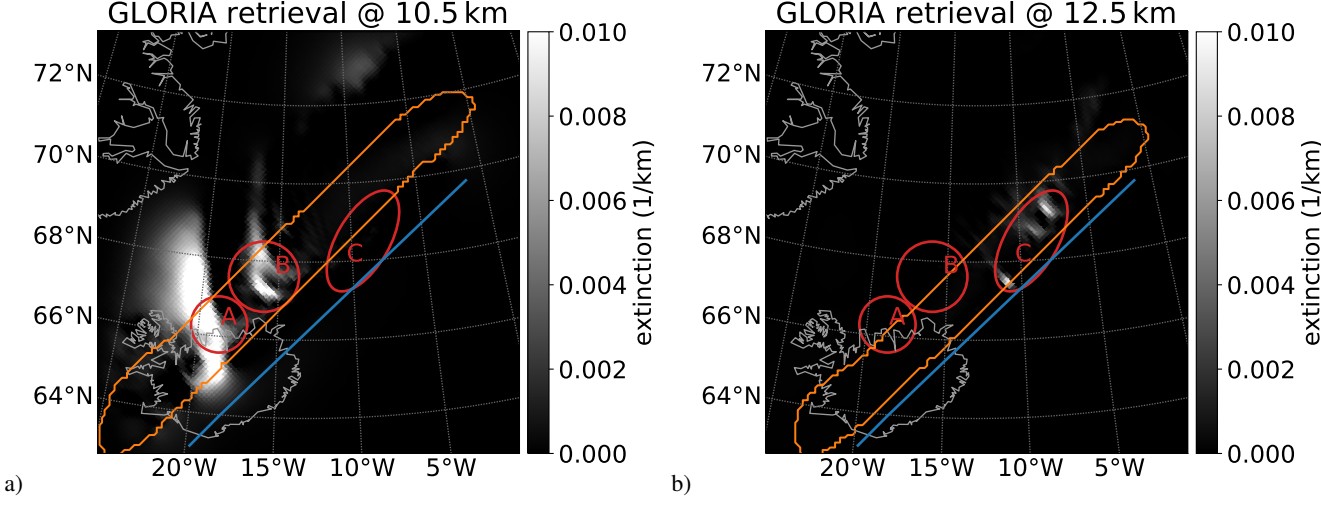

**Figure 13.** Cross-sections of extinction retrieved from GLORIA measurements. Panel (**a**) shows a horizontal cross-section at 10.5 km and panel (**b**) one at 12.5 km. Flight path is shown in blue, the trust region with high confidence in retrieval results around the tangent points of measurements is marked in orange. The three red ellipses mark areas of interest.





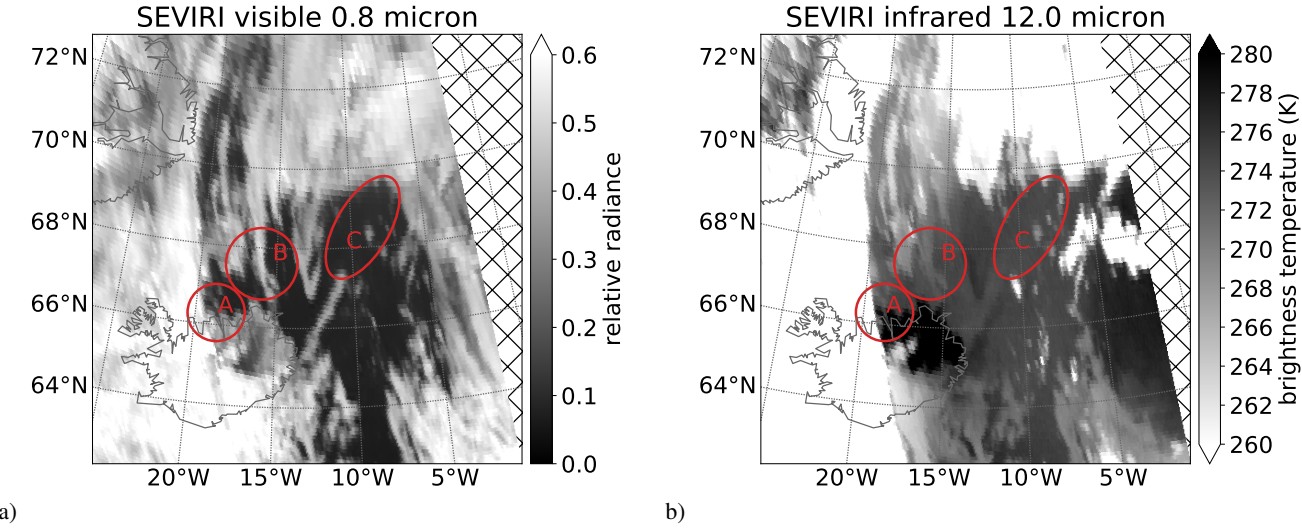

**Figure 14.** Two images taken by SEVIRI on Meteosat on 18th September 2017 at 12:00 UTC. Panel **(a)** shows the 0.8 micron visible channel in a relative scale and panel **(b)** shows the 12.0 micron infrared channel as brightness temperature. The three red ellipses mark areas of interest. Hatched areas indicate missing data.

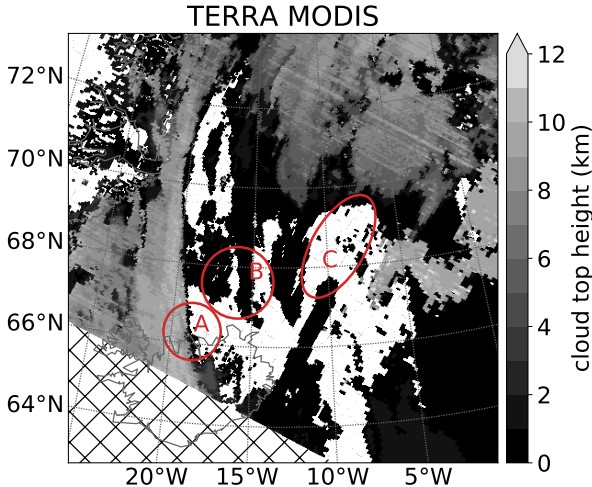

**Figure 15.** The cloud top altitude L2 product of MODIS on TERRA taken on 18th September 2017 around 13:08 UTC. White indicates no recognized cloud. The three red ellipses mark areas of interest. Hatched areas indicate missing data.