# Peer review of "Cirrus cloud shape detection by tomographic extinction retrievals from infrared limb emission sounder measurements"

_Atmospheric Measurement Techniques, 2020_

## Referee Comment (RC1) · Anonymous Referee #1 · 4 Aug 2020

**General comments**

The authors present a new method, the *convex hull cloud index*, to identify clouds in atmospheric limb measurements. As compared to the conventional cloud index method, the new algorithm shows significantly improved performance when applied to the high density measurements of a new imaging Middle InfraRed FTS limb-sounder (IRLS) that was proposed as a future Earth Explorer mission. The authors also present a second, more sophisticated and accurate, cloud detection scheme based on the 2D tomographic retrieval of cloud extinction, from limb-sounding measurements. The performance of this latter method is impressive when retrieving cloud geometrical exten-

sion and position from simulated measurements of the planned IRLS instrument. The efficiency of the method is also tested and validated on the basis of real measurements acquired by the GLORIA instrument during an aircraft campaign.

The subject of the paper is very interesting for the remote sensing scientific community and is within the scope of AMT. The methods used are sound. Apart from Section 5, which is a bit fuzzy and, to my opinion, would also require a language revision, the rest of the paper is clearly written and sufficiently concise.

I have only one main concern that, probably, can be solved by including a minimal discussion. To generate IRLS synthetic measurements, the authors assume a FTS spectral resolution 1/(2 MOPD)=1.25 cm$^{-1}$, that is more than a factor of 6 worse than the required value for this instrument to study atmospheric chemistry. Such a strong reduction of the spectral resolution makes it possible to implement a very fine spatial sampling step, as the time interval required for the individual measurements is reduced accordingly. This instrument setup is very favourable to study clouds which generally have broad spectral features and are scattered in space. On the other hand, most likely, measurements made in this configuration will not allow retrieving gas profiles with sufficient accuracy. Therefore, I feel that this instrument operating mode in practice could be commanded only rarely. The measurements the authors use for their 2D tomographic retrieval of the cloud extinction seem limited to the spectral radiances integrated in two narrow spectral intervals. Probably, these measurements could be acquired with an instrument much cheaper than the considered IRLS interferometer. Maybe, a 2 channels calibrated imager pointing at the limb could be sufficient?

In conclusion, I recommend to publish this paper in AMT, however the authors should at least include a minimal discussion regarding the above issue, and clarify / correct the text according to the specific and minor comments included below.

**Specific comments**

Lines 34–36: please note that the tomographic 2D retrieval for satellite limb measurements dates back to Carlotti et al. 2001 (I suggest to cite this paper also here, just before Steck et al. 2005). Due to technology limitations, MIPAS/ENVISAT was not an imaging interferometer. This implied that the sampling strategy had to give priority to vertical coverage and vertical sampling step rather than to the horizontal sampling. Despite of that, MIPAS limb measurements had overlapping line-of-sights, especially in the measurements acquired after 2004. Due to this characteristics of MIPAS measurements, a database of Level 2 products obtained with the 2D tomographic approach was established: see Carlotti et al. 2006 and Dinelli et al. 2010. Even if now you are getting much better results due to the improved characteristics of the planned (IRLS) and available (GLORIA) measurements, I suggest to give proper recognition also to the above mentioned work and to correct the statement at line 36, which is not true (e.g. the special MIPAS observation mode S6 was specifically designed for tomographic retrievals, see http://eodg.atm.ox.ac.uk/MIPAS/frmodes.html ).

Line 75 "to point its detector...": I guess it is the "line of sight" which is adjusted, not the detector itself.

Line 86, "15 horizontal measurement tracks covering 7∘...": do you mean across track measurements with line-of-sight spanning an angle of 7 degrees in azimuth?

Lines 85–88: see my main comment above. The configuration you assume (spectral sampling = 1.25 cm$^{-1}$ and fine spatial coverage) is certainly optimal to investigate cloud propertied which generally have broad spectral features. However, the IRLS mission had additional science objectives (gases retrieval) which, probably, cannot be

met if such a low spectral resolution is commanded. So, although feasible, I imagine this operation mode can be used only for very limited time periods. Could-you please comment / address this point?

Line 143: I believe CI is the ratio between the integrated spectral radiance in two different spectral intervals, not the "ratio between spectral regions" as mentioned here.

Lines 178–180: same comment as above (regarding lines 85–88). Please note that for CI analyses, a much cheaper instrument such as a two channels calibrated imager could be sufficient. If the IRLS instrument will be implemented on a real satellite, you may hardly ask to operate it with so low spectral resolution and fine spatial sampling just to perform the CI analysis, if the spectral resolution is degraded, a lot of information on gases spectral lines is lost.

Line 213: the computing power of current PCs varies by orders of magnitude, please give the main technical details of the platform in which you tested the algorithm.

Line 252: do you also add pseudo – random noise to the synthetic measurements? Does the measurement noise have an impact on the results presented?

Line 228: why do you retrieve the temperature profile? The ECMWF estimate is not sufficiently accurate for your purposes? For each sweep of the limb scan, the measurements included in the inversion seem limited to the two values of the integrated radiance in the two spectral intervals (centered at 796.25 cm$^{-1}$ and 835.00 cm$^{-1}$). How the number of observations compares to the number of retrieved parameters? My concern is that temperature and extinction could be very correlated in your inversion, thus in real retrievals, forward model errors (like interference errors or model errors

due to neglecting scattering) could impact strongly on the retrieved extinction, that is the target of interest to you.

Lines 238 and ff: please explain the meaning of all the symbols appearing in Eq. 1. The Levenberg–Marquardt damping, the Jacobian F', and the iteration index $i$ are not defined.

Line 238: you assume 0.1% error on the integrated radiances. Is this error figure compliant with MIPAS / IRLS NESR specifications?

Lines 242 and 243: regarding the constraint applied to extinction, it is not clear whether you constrain only the first derivative or also the actual retrieved values (usually, this would not be called regularization). Which first derivative do you constrain? The horizontals, the vertical, or all?

Line 249: presumably here you refer to the amount of RAM required. I would use the more common unit of Megabyte (MB) instead of the Mebibyte (MiB). By the way, the difference between the two is only 4.9%, thus not really relevant.

Lines 256–257: not obvious to me, I recommend to include at least a "minimal" explanation.

Line 273: did-you mean "a coarser horizontal retrieval grid"?

Figures 2 to 6: could you specify somewhere, where is the satellite instrument located in these figures? I guess the satellite is on the left, given the difficulty to detect exactly

the Cloud Top on the right side of the more opaque clouds.

Figures 2 to 7: the top title is not very informative for the readers, it could be removed.

Lines 338–339: in panel (e) of Figure 9, I would have expected to see the retrieved extinctions, however the text is not very explicit on this point. Moreover, the caption of Figure 9 states that panel (e) shows the extinction values "used for generating the simulated measurements". This is confusing, could-you please clarify what you are really showing?

Lines 341–347: this is strange again. From the left plot of panel (d) on Fig. 9, I see that the CI determined on the basis of the convex hull, results with cloudy pixels at 5 km at latitudes of 45–50 degrees North. However, looking at panel (b) of Fig. 10, it does not seem that the cloud system extends down to 5 km. Again, here I don't understand why in Fig. 10b you don't plot the 0.001 $km^{-1}$ contour surface of the retrieved extinction, which should give the best performance. Did I misunderstand something? Please clarify.

Lines 359–362: why do you change the retrieval approach in the case of GLORIA measurements? Specifically, why the ECMWF temperature is considered sufficiently accurate for this case and not enough accurate in the test simulations presented earlier? I agree that if you do not retrieve temperature, the microwindow centered about 796.25 $cm^{-1}$ is no longer required, however, you also change the second microwindow that in earlier tests was centered about 835.00 $cm^{-1}$. Why? Please justify. Please, justify also the choice of ignoring gases concentrations in this retrieval.

Lines 364–365: It seems that two cameras were operating together with GLORIA in

this flight. Please introduce earlier all the instruments used and define also the main features of the cameras, e.g. which is their spectral band-pass?

Lines 365–368: not clear at all. Maybe it will be more clear when the operation mode of the two cameras will be better explained.

Line 374: the inversion problem is said to be "ill-conditioned" (and this is especially true because relatively few measurements were used in the inversion, as compared to the amount of retrieval parameters).

Line 379: please specify how you define the "well resolved region". How many degrees of freedom (DOF) do you have, for each retrieval grid point, in the well resolved region?

Figure 11: please add labels in the horizontal and vertical axes. In the figure caption, I guess you meant "visible" in place of "visual". The top title does not provide useful information to the reader.

Line 383: which instrument measured the brightness temperatures reported in Fig. 12? GLORIA? To which wavenumber or spectral band do the reported brightness temperatures refer?

Lines 384–393: the retrieval or inversion of measurements is the "rigorous" approach while, I guess, this is a qualitative interpretation of auxiliary / additional measurements that were not adequately introduced.

Lines 395–400: I think a Figure with some maps reporting the number of DOFs per

retrieval grid point at some fixed altitudes would be more illustrative and clear than this qualitative explanation.

Line 442: extinction is in km$^{-1}$. From Fig. 13, the extinction of cloud B seems of the order of 0.01 km$^{-1}$. Please explain how you infer the extinction of 0.1 km$^{-1}$.

**Minor comments**

Figure 1 caption: profile / profiles ==> limb scan(s)

Lines 236–237: all symbols of the equation are either vectors or matrices, therefore I would not use italic fonts. Unlike the others, the last "F" is italic.

Line 233: for "F" I would use bold, roman fonts everywhere.

Line 267: "size".

Line 278: "finer spatial sampling"

Line 282: next ==> nearly

Line 332: structures

Line 387: "shrunken" ??

Line 485: "please note that"

**References**

Massimo Carlotti, Gabriele Brizzi, Enzo Papandrea, Marco Prevedelli, Marco Ridolfi, Bianca Maria Dinelli, and Luca Magnani, "GMTR: Two-dimensional geo-fit multitarget retrieval model for Michelson Interferometer for Passive Atmospheric Sounding/Environmental Satellite observations," Appl. Opt. 45, 716-727 (2006).

Dinelli, B. M., Arnone, E., Brizzi, G., Carlotti, M., Castelli, E., Magnani, L., Papandrea, E., Prevedelli, M., and Ridolfi, M.: The MIPAS2D database of MIPAS/ENVISAT measurements retrieved with a multi-target 2-dimensional tomographic approach, Atmos. Meas. Tech., 3, 355–374, https://doi.org/10.5194/amt-3-355-2010, 2010.

---

## Referee Comment (RC2) · Anonymous Referee #2 · 10 Sep 2020

General comments:

Passive limb sounding provides a very sensitive method to detect clouds from space. An improved algorithm, the convex hull cloud index, is presented which allows better performance than traditional cloud index methods when applied to instruments such as the IRLS concept with operational scanning modes capable of providing sufficient oversampling of airmasses along the line of sight. A 2D tomographic retrieval of cloud extinction is also presented that is applicable to IRLS type instruments and is validated with measurements from the airborne GLORIA instrument.

I have no major concerns to address and the paper would be suitable for publication in

[Figure]

AMT following attention to some minor revisions.

Specific comments:

/xxx/ means delete xxx

[xxx] means add xxx

L4: "is better in detecting the horizontal cloud extent". Why not say "has improved horizontal resolution"?

L25: suggest "rigorous" instead of "proper"

L26: most sensitive [passive] method ... /are/ [is]

L31: suggest "the poor" rather than "a bad"

L34-L37: Some earlier work on 2D retrievals is omitted in this discussion e.g. as cited in Ungermann (2013) ... see p16... "This is in principal similar to the concept of tomographic 2-D retrievals of satellite limb sounder data, which were first produced by Carlotti et al. (2001) and Steck et al. (2005) for the Michelson Interferometer for Passive Atmospheric Sounding (MIPAS) and by Livesey and Read (2000) for the Microwave Limb Sounder."

Massimo Carlotti, Bianca Maria Dinelli, Piera Raspollini, and Marco Ridolfi, "Geo-fit approach to the analysis of limb-scanning satellite measurements," Appl. Opt. 40, 1872-1885 (2001)

Nathaniel J. Livesey and William G. Read, "Direct Retrieval of Line-of-Sight Atmospheric Structure from Limb Sounding Observations", GEOPHYSICAL RESEARCH LETTERS, VOL. 27, NO. 6, PAGES 891-894, MARCH 15, 2000

L36: and many other instances: line-of-sights => lines-of-sight

L61-L62: Wasn't most of the MIPAS mission carried out with the reduced resolution mode?

L62: Which MIPAS operational mode does this 1.5km vertical resolution correspond to?

L75: suggest "relative" rather than "compared"

L76: airmasses [at an angle]

L82: is [a concept for] an

L86: a spectral sampling of 1.25cm-1 seems very large

L86: is this +/- 7 deg or +/- 3.5 deg in azimuth?

L104: CLaMS is not defined until L106

L128-L129: This is the only mention of aerosols in the paper. Are aerosols assumed to not exist in the simulations shown here? Are their effects assumed negligible and if so to what extent? What about elevated sulfate aerosols in volcanic plumes? What about the potential for aerosol-cloud interaction studies?

L138: It's not clear here – would the FTS produce data sampled only at the coarse resolution? Is the spectral resolution sufficient to do a good enough job with the gas retrievals? I'm assuming (from L125) that the emissivity growth approximation look-up tables are generated using high resolution line-by-line calculations and spectrally integrated to the coarse resolution. Is that the case? Some more detailed explanation seems necessary.

L145-L153: Please provide some information on the approximate optical depths corresponding to the CI values 1.1 and 4 and therefore what you mean by thin and thick clouds.

L158: suggest "usefully" rather than "meaningfully"

L172: Maybe explain the concept of a "convex hull"

L174: Maybe clarify this a bit better. What I am getting from this is that "below the

threshold" means that the CI detection method got triggered. However, the CI detects a spurious cloud at the tangent, but it was triggered by a real cloud well above the tangent.

L178: this aspect of [the] different

L181: "A common approximation of the spatial origin of a radiance measurement is assigning it to its tangent point, which is the location along its line-of-sight that is closest to the surface."

The radiance measured is the result of the integration along the refracted ray path (which may also involve scattering of radiation into the line-of-sight). The tangent height is just a convenient label to use.

L185: extinction cross-section/s/

L193: Especially [optically] thin

L197-L210: In this section (and everywhere else) it's not clear whether unrefracted or refracted tangent rays are being described. Refraction will cause the actual tangent point to be lowered and move closer to the instrument compared to the geometric (unrefracted) tangent point.

L216: The first reference to the CI being a "color-ratio" probably should be on L143.

L237:Eqn 1: The role of lambda_i is not defined. F' is not mentioned.

L245: to allow /for/ oscillations

L250: 1/2 orbit in 25 mins or 1 day (typically 14.5 orbits) in 6 hrs.

L284: What is the optical depth cut-off value?

L361: What is the clear-sky optical depth for the lowest tangent height you are considering?

L383: Brightness temperatures?

L392: I don't see how the retrieval could distinguish between spatial vs temporal evolution.

L410: is /in/ [on] the order ... in [the] flight

L426: What about evaporation?

L437: in [a] north-south

L441 in [the] nadir]

L442: units typo ... 0.1 km[-1]

─────────────────────────────

---

## Author Comment (AC1) · 16 Oct 2020

Dear sirs and madams,

please find attached our author reply.

Best regards, Jörn Ungermann

Please also note the supplement to this comment: https://amt.copernicus.org/preprints/amt-2020-256/amt-2020-256-AC1-supplement.pdf

---

## Author Response (AR1)

We thank the reviewers for their many very helpful comments and insightful suggestions. The plots and discussions were improved and many descriptions are now more precise.

We do not repeat minor technical suggestions, e.g., for wording, unless we did *not* apply them straightforwardly. Attached in the end is the revised paper with additions/deletions marked in red and blue colours.

**1 Reply to Referee #1**

**1.1 Major Comments**

1. *I have only one main concern that, probably, can be solved by including a minimal discussion. To generate IRLS synthetic measurements, the authors assume a FTS spectral resolution 1/(2 MOPD)=1.25 cm$^{-1}$, that is more than a factor of 6 worse than the required value for this instrument to study atmospheric chemistry. Such a strong reduction of the spectral resolution makes it possible to implement a very fine spatial sampling step, as the time interval required for the individual measurements is reduced accordingly. This instrument setup is very favourable to study clouds which generally have broad spectral features and are scattered in space. On the other hand, most likely, measurements made in this configuration will not allow retrieving gas profiles with sufficient accuracy. Therefore, I feel that this instrument operating mode in practice could be commanded only rarely. The measurements the authors use for their 2D tomographic retrieval of the cloud extinction seem limited to the spectral radiances integrated in two narrow spectral intervals. Probably, these measurements could be acquired with an instrument much cheaper than the considered IRLS interferometer. Maybe, a 2 channels calibrated imager pointing at the limb could be sufficient?*

   *In conclusion, I recommend to publish this paper in AMT, however the authors should at least include a minimal discussion regarding the above issue, and clarify / correct the text according to the specific and minor comments included below.*

   *Lines 85–88: see my main comment above. The configuration you assume (spectral sampling = 1.25 cm$^{-1}$ and fine spatial coverage) is certainly optimal to investigate cloud propertied which generally have broad spectral features. However, the IRLS mission had additional science objectives (gases retrieval) which, probably, cannot be met if such a low spectral resolution is commanded. So, although feasible, I imagine this operation mode can be used only for very limited time periods. Could you please comment / address this point?*

   *Lines 178–180: same comment as above (regarding lines 85–88). Please note that for CI analyses, a much cheaper instrument such as a two channels calibrated imager could be sufficient. If the IRLS instrument will be implemented on a real satellite, you may hardly ask to operate it with so low spectral resolution and fine spatial sampling just to perform the CI analysis, if the spectral resolution is degraded, a lot of information on gases spectral lines is lost.*

   The instrument was designed with two major operating modes in mind, one exploring the atmospheric dynamics (major circulation systems, gravity waves, UTLS exchange by means of water vapour, CFCs and ozone) modelled more after the CRISTA satellite and one dedicated to examining the chemical composition of the atmosphere modelled more after the (nominal operating mode of the) MIPAS-ENVISAT instrument. Both operating modes were always seen as equally important and neither was a "special, seldom to be used" mode. The exact way to split the observation time was not finally settled, but the following was noted in the report for assessment: *The typical operation time of one mode is between one orbit and one week or even more. The baseline is for observing times to be comparable in the two modes..* This equal split should give a sufficient amount of observations for proper statistical

evaluation of data from both modes (e.g., gravity wave occurrence and chemical composition). The quantities of temperature, water vapour, ozone, and CFCs would be continuously available during both modes (albeit with maybe different spatial characteristics).

Please note also that many more trace gases were derived from measurements of the CRISTA spectrometer, which also had a spectral resolution much coarser than the IRLS CM mode. While this was not investigated in detail, it is plausible that more than the above listed trace gases could be derived from DM mode measurements ( although not the full range of those available from scientific processors evaluating MIPAS-ENVISAT measurements, obviously).

For an instrument solely dedicated to gravity waves (or cloud shape retrieval for that matter) a simpler radiometer might be sufficient, even though it would be certainly less accurate (the resolved spectrum allows to better remove emissions of background gases and better overall calibration). However, the scientific requirements for the IRLS to derive water vapour and ozone highly spatially resolved in the UTLS makes a FTS already much more attractive, the possibilities of exploiting the remainder of the spectrum for other objectives aside. A HIRDLS-like successor instrument could likely fulfill most of the requirements of the IRLS dynamics mode, though.

We added to the IRLS instrument description: *This corresponds to the 'dynamics mode', which was envisioned to be used for about half of the instrument measurement time; its primary purpose was a high spatial resolution to reveal processes associated with mixing and convective outflow in the UTLS, as well as three-dimensionally resolving gravity waves to determine momentum fluxes driving global circulation systems (ESA, 2012).*

**1.2 Specific Comments**

1. *Lines 34–36: please note that the tomographic 2D retrieval for satellite limb measurements dates back to Carlotti et al. 2001 (I suggest to cite this paper also here, just before Steck et al. 2005). Due to technology limitations, MIPAS/ENVISAT was not an imaging interferometer. This implied that the sampling strategy had to give priority to vertical coverage and vertical sampling step rather than to the horizontal sampling. Despite of that, MIPAS limb measurements had overlapping line-of-sights, especially in the measurements acquired after 2004. Due to this characteristics of MIPAS measurements, a database of Level 2 products obtained with the 2D tomographic approach was established: see Carlotti et al. 2006 and Dinelli et al. 2010. Even if now you are getting much better results due to the improved characteristics of the planned (IRLS) and available (GLORIA) measurements, I suggest to give proper recognition also to the above mentioned work and to correct the statement at line 36, which is not true (e.g. the special MIPAS observation mode S6 was specifically designed for tomographic retrievals, see http://eodg.atm.ox.ac.uk/MIPAS/frmodes.html ).*

We will correct the text and also cite the Carlotti paper, which predates the given cites by quite a bit.

The L2 products of MIPAS, as those of other conventional limb sounders, improve greatly by using tomographic measurement techniques. However, the main advantage for a tomographic retrieval of MIPAS data is not an increased spatial resolution, but the increased reliability against gradients in trace gases and temperature along the line-of-sight (a *very* important feature for limb sounders, but not the subject of this particular study), not the spatial horizontal resolution. And while lines-of-sight overlap, they do so only in "higher" layers, not close to the tangent point layer. The special tomographic measurement mode of MIPAS, with a horizontal spacing of 180 km, would indeed increase the spatial resolution compared to the figures of 200 to 500 km for the horizontal resolution of the nominal operating mode given by von Clarmann et al. (2009), and it served well as a proof of concept for this capability. We adopted the text:

*,,While tomographic retrievals have become state of the art for limb sounders in general (e.g. Livesey and Read, 2000; Carlotti et al., 2001; Steck et al., 2005; Livesey et al., 2006; Christensen et al., 2015), in-orbit instruments do not oversample extensively; e.g., the MIPAS instrument on ENVISAT (Fischer et al., 2008) was operated to have in nominal measurement modes non-overlapping lines-of-sight in the tangent layer. Tomography was, due to instrument limitations, mostly performed to increase the retrieval accuracy in the presence of gradients in retrieved quantities along the line-of-sight."*

2. *Line 75 "to point its detector...": I guess it is the "line of sight" which is adjusted, not the detector itself.*

GLORIA does not use mirrors for its pointing, but actually moves the whole spectrometer, including the detector, in a 3-D gimbal mount. To avoid confusion, we changed the text to *,,to point the instrument"*.

3. *Line 86, "15 horizontal measurement tracks covering 7°": do you mean across track measurements with line-of-sight spanning an angle of 7 degrees in azimuth?*

We adopted the text to state *,,±3.5°"*.

4. *Line 143: I believe CI is the ratio between the integrated spectral radiance in two different spectral intervals, not the "ratio between spectral regions" as mentioned here.*

The suggestion is indeed more precise. We updated the definition to *,,The CI is defined as the ratio between the radiance averaged over a spectral region with a strong emission feature, such as the $CO_2$ Q-branch at 12.6 $\mu$m, and the radiance averaged over an atmospheric window, such as the one located at 12 $\mu$m. "*

5. *Line 213: the computing power of current PCs varies by orders of magnitude, please give the main technical details of the platform in which you tested the algorithm.*

The system was a *,,AMD EPYC 7351P 16-Core Processor operating at 2.9* GHz*"*, which was amended in the text.

6. *Line 252: do you also add pseudo – random noise to the synthetic measurements? Does the measurement noise have an impact on the results presented?*

We included consistently 0.8 nW/(cm$^2$srcm$^{-1}$) corresponding to the envisioned noise level of the IRLS instrument for microwindows of the given width. But we found this noise to be insignificant for the task at hand. We did not use a different noise specification for MIPAS to stay consistent, especially as no effect was noticeable. The focus is to compare the relative spatial detection capabilities and not the detection thresholds.

We added *,,To focus on the relative capabilities of the algorithms, we added Gaussian noise to simulated MIPAS or IRLS measurements with a standard deviation of 0.8* nW/(cm$^2$srcm$^{-1}$)*."*

7. *Line 228: why do you retrieve the temperature profile? The ECMWF estimate is not sufficiently accurate for your purposes? For each sweep of the limb scan, the measurements included in the inversion seem limited to the two values of the integrated radiance in the two spectral intervals (centered at 796.25 cm$^{-1}$ and 835.00 cm$^{-1}$). How the number of observations compares to the number of retrieved parameters? My concern is that temperature and extinction could be very correlated in your inversion, thus in real retrievals, forward model errors (like interference errors or model errors due to neglecting scattering) could impact strongly on the retrieved extinction, that is the target of interest to you.*

We assume that ECMWF temperature is typically known well enough. Except for extreme circumstances, such a heavy small-scale gravity wave activity, we seldom found ECMWF

off by more than $3\,\mathrm{K}$ in recent aircraft campaigns. However, with 2-D tomography one is typically able to derive temperature without too much hassle such that one does not need to rely on assumptions (always preferable). When not retrieving temperature, we'd have needed to model the assumed discrepancies of ECMWF and estimate the effect on extinction, strictly because of the correlation between them. We believed the setup to be more realistic and difficult by including the temperature in the retrieval study.

For the 3-D GLORIA example, we assumed ECMWF to be sufficiently correct to drastically simplify the very costly, and also more unstable, 3-D retrieval.

For the satellite extinction retrievals, there are typically twice as many unknowns as measurements. This is largely due to atmospheric state parameters in boundary regions (above, below, left, right) and some minor over-sampling.

We also believe the listed error sources to impact the derived extinction value. Especially scattering can cause an error in the double-digit percentage range. It causes, typically, an overestimate in the derived extinction, as warmer radiances from lower layers or the ground are scattered into the line-of-sight. As such, scattering improves the cloud detection capability. Please note that scattering also impacts cloud detection by colour ratios.

An error in temperature should have, generally, also a small effect, percentage wise. The signal generated by clouds is typically so large that even large errors will not cause us to "miss" the cloud.

Evaluating the derived extinction quantitatively to derive cloud parameters such as ice water path, etc., would require a more careful approach.

8. *Lines 238 and ff: please explain the meaning of all the symbols appearing in Eq. 1. The Levenberg–Marquardt damping, the Jacobian F', and the iteration index i are not defined.*

We amended the section to include the missing information.

9. *Line 238: you assume 0.1% error on the integrated radiances. Is this error figure compliant with MIPAS / IRLS NESR specifications?*

No, it isn't. The noise added to the measurements was defined according to the IRLS specification (see above), but it is insignificant here compared to the relative error caused due to the use of different grids for generating the synthetic measurements and performing the retrieval. We added: *,,The matrix $\mathbf{S}_\epsilon$ is set up assuming a 0.1 % error in radiance, which has been chosen to allow for differences caused by the use of different grids for generating the synthetic measurements and the retrieval itself."*

10. *Lines 242 and 243: regarding the constraint applied to extinction, it is not clear whether you constrain only the first derivative or also the actual retrieved values (usually, this would not be called regularization). Which first derivative do you constrain? The horizontals, the vertical, or all?*

We rephrased more specifically: *For extinction, the first derivatives in both spatial directions are constrained using the same correlation lengths in addition to imposing a weak constraint of the absolute extinction values towards the zero profile.*

11. *Line 249: presumably here you refer to the amount of RAM required. I would use the more common unit of Megabyte (MB) instead of the Mebibyte (MiB). By the way, the difference between the two is only 4.9%, thus not really relevant.*

We changed the unit, which, due to rounding, does not matter, indeed.

12. *Lines 256–257: not obvious to me, I recommend to include at least a "minimal" explanation.*

We added a reference to the figure and an explanation as *,,The 'pixels' corresponding to MIPAS measurements in Fig. 2b are very coarse compared to the fine structure of the clouds*

*contained in the CALIOP data due to the much sparser horizontal sampling density of the MIPAS instrument."*

13. *Line 273: did-you mean "a coarser horizontal retrieval grid"?*

Yes. We changed the text accordingly.

14. *Figures 2 to 6: could you specify somewhere, where is the satellite instrument located in these figures? I guess the satellite is on the left, given the difficulty to detect exactly the Cloud Top on the right side of the more opaque clouds.*

The satellite is not located at one position only, as this is a tomographic measurement, i.e., it combines the measurements of the satellite from multiple positions. We added the sentence ,,*The satellite looks northwards in these simulations.*" to the captions of these figures.

15. *Figures 2 to 7: the top title is not very informative for the readers, it could be removed.*

We complied and added the information into the caption, where we found this necessary.

16. *Lines 338–339: in panel (e) of Figure 9, I would have expected to see the retrieved extinctions, however the text is not very explicit on this point. Moreover, the caption of Figure 9 states that panel (e) shows the extinction values "used for generating the simulated measurements". This is confusing, could-you please clarify what you are really showing?*

To be more consistent with the previous figures, we moved the row e) to the top. This is indeed the extinction used to perform the simulations. As mentioned in the main text, we only employed the CI based methods on the CLaMS 3-D data. We also changed the cloud-shape result to the same white-blue colour scale used in the previous plots.

*Lines 341–347: this is strange again. From the left plot of panel (d) on Fig. 9, I see that the CI determined on the basis of the convex hull, results with cloudy pixels at 5 km at latitudes of 45–50 degrees North. However, looking at panel (b) of Fig. 10, it does not seem that the cloud system extends down to 5 km. Again, here I don't understand why in Fig. 10b you don't plot the 0.001 $km^{-1}$ contour surface of the retrieved extinction, which should give the best performance. Did I misunderstand something? Please clarify.*

This confusion is the result of the unfortunate placement of the "true extinction" in the preceding plot, which was corrected. This 3-D CLaMS section focuses on the convex hull algorithm for cloud shape retrievals and not the extinction retrieval. The focus of this section is less to compare the different methods, but to demonstrate the 3-D capabilities. We also assume that a tomographic extinction retrieval (2-D or even 3-D) would deliver slightly superior results as before, but did not actually set it up.

17. *Lines 359–362: why do you change the retrieval approach in the case of GLO-RIA measurements? Specifically, why the ECMWF temperature is considered sufficiently accurate for this case and not enough accurate in the test simulations presented ear- lier? I agree that if you do not retrieve temperature, the microwindow centered about 796.25 $cm^{-1}$ is no longer required, however, you also change the second microwindow that in earlier tests was centered about 835.00 $cm^{-1}$. Why? Please justify. Please, justify also the choice of ignoring gases concentrations in this retrieval.*

Due to historical different use of cloud index spectral regions between satellite and airborne instruments, there were some inconsistencies between the satellite simulations and the GLO-RIA retrieval. We consolidated the microwindow region employed for GLORIA data to 831.2 to 835.0 (still slightly different due to the given spectral sampling of 0.2 wavenumbers in the

level 1 data, but as close as feasible). This microwindow is now also consistently used for the retrieval and all depictions of GLORIA IR images and brightness temperatures. The retrieval results changed only insignificantly. We explained the reasoning for discarding the temperature retrieval and CO2-Q-branch window better in the main text: *,,We used all images taken between 11:10 UTC and 12:35 UTC in the reconstruction. The 3-D retrieval is computationally significantly more expensive compared to the 2-D retrievals as the number of unknowns is much higher and the involved algorithms scale with a power of given unknowns. The atmospheric volume is also much less constrained by the measurements as the number of unknowns vastly outnumbers the measurements and the distribution of information is far from homogeneous. We found that deriving temperature and extinction similar to the 2-D setup works well within the volume covered by tangent points, but quickly deteriorates outside. Thus, we neglect the temperature retrieval here, as this improved the retrieval on three fronts. First, this allows discarding of the 792 $cm^{-1}$ window and to not take any trace gas emissions into account in the forward modelling, drastically improving the forward model speed by orders of magnitude. Second, it halved the number of unknowns, thereby increasing convergence speed of iterative solvers by a factor of roughly four and decreasing memory consumption by half. Third, this also stabilized the extinction retrieval such that the extinction values outside the core volume are less affected by retrieval artifacts. The employed single microwindow averaged all spectral samples between 831.2 $cm^{-1}$ and 835.0 $cm^{-1}$ (GLORIA operated in a mode that allows for a spectral sampling of 0.2 $cm^{-1}$ during this portion of the flight)."*

18. *Lines 364–365: It seems that two cameras were operating together with GLORIA in this flight. Please introduce earlier all the instruments used and define also the main features of the cameras, e.g. which is their spectral band-pass?*

 *Lines 365–368: not clear at all. Maybe it will be more clear when the operation mode of the two cameras will be better explained.*

There are just two 'cameras' contained in GLORIA: the main camera is the IR spectrometer, which has been described in detail, while additionally a "standard" camera in the visible range of the spectrum has been mounted; this secondary camera has several technical issues that make it difficult to use the data scientifically except for getting a better overview of an imaged situation. Information about the passband characteristics of the optics or the sensitivity of the RGB detector pixels are not available and there were so far also no calibration efforts. It has been used so far to get an overview of the imaged situation in a larger context due to the wider FOV.

We added *,,In addition to the IR instrument, there is also a standard camera with three channels (red/green/blue) operating in the visible range that observes a much wider field-of-view"* to the instrument description.

We also updated the caption of the image to *,,A superposition of a visible and infrared image taken at 11:50:31 UTC showing a cirrus cloud located at 10.5 km. The visible image has a much wider field-of-view than the infrared one. The infrared image has been extracted from the spectrally resolved GLORIA measurements and shows the averaged radiance over the spectral range from 831.2 $cm^{-1}$ to 835.0 $cm^{-1}$; it has also been shifted to the right for better comparability and the radiance is depicted in a logarithmic scale with arbitrary units;"* to clarify that the IR image has not been taken by a third instrument, but has been taken by GLORIA (it is an imager after all).

19. *Line 374: the inversion problem is said to be "ill-conditioned" (and this is especially true because relatively few measurements were used in the inversion, as compared to the amount of retrieval parameters).*

The inverse problem is under-determined and has thus a non-trivial null-space. Thus, it is certainly ill-conditioned. It is ill-posed as well. The approximation of the original problem by the regularized problem takes care of both issues.

We expanded the text *,, This number is significantly larger than the number of measurements, making this a drastically under-determined problem.*

*The inverse problem, i.e., identifying an atmospheric state fitting to the measurements, is an ill-posed problem. To solve it, we approximate it by a well-posed, regularized formulation. To regularize the problem, we employ constraints of zero-th and first order. "*

20. *Line 379: please specify how you define the "well resolved region". How many degrees of freedom (DOF) do you have, for each retrieval grid point, in the well resolved region?*

    *Lines 395–400: I think a Figure with some maps reporting the number of DOFs per retrieval grid point at some fixed altitudes would be more illustrative and clear than this qualitative explanation.*

Such a plot would be obviously very useful, but is too expensive to compute for the given, highly resolved setup. For other setups, we found the region withing the volume covered by tangent points to have a consistent, good resolution, while the shape of the AVK outside becomes quickly very irregular making already the definition of a FWHM resolution difficult (Ungermann et al., 2011; Ungermann, 2011; Krisch et al., 2018).

Computing a row of the AVK is only slightly one order of magnitude less expensive as the full retrieval and any cross-section has ten-thousands of points. But we computed the degrees of freedom for selected representative points inside and outside the well-resolved volume. Inside the well-resolved volume typical values are in the order of 1/20, i.e. about 20 samples share one piece of information. This value is to be expected with the oversampling of the atmosphere. Outside, it drops to about an order of magnitude less.

Accessing the trace of the AVK is sadly infeasible as well; at least we have not found a way to access it for retrievals of this size, yet.

A full diagnostic is only feasible for retrievals with a few ten thousands of unknowns due to the computational effort and numerical instability. 2-D satellite retrievals are at the border of this figure, but can be typically set up in a way to be smaller.

We added the DOF figure as *,, Within the well-resolved volume the analysis shows an information content of about 0.05, i.e. 20 samples share about one degree of freedom; outside it drops to an order of magnitude less, but numerical instabilities of the involved equation systems makes the latter difficult to compute precisely. ".* Resolution as FWHM in km was already specified before.

21. *Figure 11: please add labels in the horizontal and vertical axes. In the figure caption, I guess you meant "visible" in place of "visual". The top title does not provide useful information to the reader.*

We added the labels, the title, and replaced visual with visible.

22. *Line 383: which instrument measured the brightness temperatures reported in Fig. 12? GLORIA? To which wavenumber or spectral band do the reported brightness temperatures refer?*

The brightness temperature was measured by the GLORIA IR spectrometer. The spectral range was added as: *,, Brightness temperature of GLORIA measurements averaged over the wavenumber range from $831.2\,\mathrm{cm}^{-1}$ to $835\,\mathrm{cm}^{-1}$ (the same that is used as in the tomographic retrieval) over time sorted according to azimuth angle in relation to aircraft heading. "*

23. *Lines 384–393: the retrieval or inversion of measurements is the "rigorous" approach while, I guess, this is a qualitative interpretation of auxiliary / additional measurements that were not adequately introduced.*

This is a qualitative interpretation of the actual measurements being used in the retrieval. The brightness temperatures depicted were derived from the GLORIA measurements used

in the retrieval (which has been made clearer in the caption). Only the retrieval is able to do a quantitative and optimal interpretation (rigorous approach), but an intuitive approach also works up to a certain degree and is useful to determine expectations with respect to retrieval outcome and help in interpretation thereof.

We clarified this as *,,Figure 12 shows the measured brightness temperatures of the averaged microwindow used in the retrieval taken at different azimuth angles. Just these three given angles give already some insight into the real cloud structure (while the retrieval has also access to the full set of angles)."*

24. *Line 442: extinction is in $km^{-1}$ . From Fig. 13, the extinction of cloud B seems of the order of 0.01 $km^{-1}$ . Please explain how you infer the extinction of 0.1 $km^{-1}$*

There were a couple of issues with the numbers used in this estimate.

Looking at the underlying data, the brightness temperature surrounding the observed cloud towards East is $274.3\pm0.5$K. The brightness temperature of the cloud in the region imaged by GLORIA is $272.7\pm0.5$K. The cloud itself is slightly thicker than $1\,$km, between 1.25 and $1.5\,$km. The extinction varies also over it with values of up to $0.017\,$km$^{-1}$. (and here, a zero was indeed missing!). We adopted the colour scale of the GLORIA result plot to make the higher values more visible.

These values can be used for a rough estimate that results in a smaller transmissivity of a vertical ray through the cloud of $\exp(- \approx 1.5\cdot \approx 0.014) \approx 0.98$. Applying this transmittance to the ground level brightness temperature $(274.3\,\mathrm{K})$ causes a drop in brightness temperature by about $1.3\,$K to $273.0\,$K neglecting emission/scattering by the (comparatively cold) cloud itself.

We modified the paragraph to *This cloud is $\approx1.5\,$km thick with an extinction of $\approx0.014\,$km$^{-1}$ and hence, should reduce the measured nadir brightness temperature by $\approx1.3\,$K, which is roughly consistent with the difference of 1.5 K observed by SEVIRI between the cloud and surrounding air.*

25. *Lines 236–237: all symbols of the equation are either vectors or matrices, therefore I would not use italic fonts. Unlike the others, the last "F" is italic. Line 233: for "F" I would use bold, roman fonts everywhere.*

We follow the guideline of AMT to use bold letters for matrices and bold italics for vectors. We interpret this to be applicable to matrix- and vector-*valued* functions as well. As such $\boldsymbol{F}$ is formatted with 'vec' while $\mathbf{F}'$ is formatted with 'mat'.

**2 Reply to Referee #2**

**2.1 Comments**

1. *L34-L37: Some earlier work on 2D retrievals is omitted in this discussion e.g. as cited in Ungermann (2013) ... see p16 ... "This is in principal similar to the concept of tomographic 2-D retrievals of satellite limb sounder data, which were first produced by Carlotti et al. (2001) and Steck et al. (2005) for the Michelson Interferometer for Passive Atmospheric Sounding (MIPAS) and by Livesey and Read (2000) for the Microwave Limb Sounder."*

   - *Massimo Carlotti, Bianca Maria Dinelli, Piera Raspollini, and Marco Ridolfi, "Geo-fit approach to the analysis of limb-scanning satellite measurements," Appl. Opt. 40, 1872-1885 (2001)*
   - *Nathaniel J. Livesey and William G. Read, "Direct Retrieval of Line-of-Sight Atmo- spheric Structure from Limb Sounding Observations", GEOPHYSICAL RESEARCH LETTERS, VOL. 27, NO. 6, PAGES 891-894, MARCH 15, 2000*

The list was by no means intended to be comprehensive. We agree, that the listed papers predate the given references by a couple of years and should thus be added.

2.  *L61-L62: Wasn't most of the MIPAS mission carried out with the reduced resolution mode?*
    *L62: Which MIPAS operational mode does this 1.5km vertical resolution correspond to?*

Yes. We enhanced the description slightly: ,,*We employ here the reduced (or optimised) spectral resolution of 0.0625* $cm^{-1}$ *that was used in the majority of its operational years. Similarly, we employ the vertical sampling of* $\approx$*1.5* km *in the UTLS being mostly used from 2005 till the end of operation in 2012 (RR27/nominal mode).* "

3.  *L86: a spectral sampling of 1.25$cm^{-1}$ seems very large*

The original PREMIER IRLS was assumed to operate in two different modes; a chemistry mode with a spectral sampling of 0.2 $cm^{-1}$ and a dynamics mode with a spectral sampling of 1.25 $cm^{-1}$ (the spectral resolution after apodization is even worse). The spectral sampling (i.e. interferogram length) was reduced for dynamics mode to enable a higher spatial resolution at the same downlink data rate (which was limited for a variety of reasons). The major primary scientific interest for the dynamics mode were the detection of gravity waves (by means of temperature) and stratosphere-troposphere exchange (CFC-11, O3, and H2O); all these targets could be successfully derived in sophisticated retrieval studies. While this coarse resolution does not allow the resolving of individual lines, it still can be thought of as a highly configurable, and thus much more versatile radiometer.

See also discussion of the first point of referee #1.

4.  *L86: is this +/- 7 deg or +/- 3.5 deg in azimuth?*

We adopted the text to state ,,$\pm 3.5°$ ".

5.  *L128-L129: This is the only mention of aerosols in the paper. Are aerosols assumed to not exist in the simulations shown here? Are their effects assumed negligible and if so to what extent? What about elevated sulfate aerosols in volcanic plumes? What about the potential for aerosol-cloud interaction studies?*

This paper is only concerned with ice clouds, which are the most common small-scale structure with very strong emissions in the UTLS. Other work by some of the authors deals with enhancements of the CI to an Aerosol/Cloud Index for the detection of sulfate aerosol and ash. Such work can, obviously, be immediately combined with the convex hull method. A tomographic extinction retrieval can also be extended in a similar manner by adding more spectral samples in atmospheric windows to distinguish between emitters of different spectral signature.

We focus here on the spatial aspect though. Future work might explore how to derive more information such as microphysical cloud characteristics or different emitters such as nitric acid trihydrate (NAT) PSCs, ash, sulfate aerosol, or dust.

We added a paragraph noting the limitations of the study: ,,*The cloud index is also sensitive to an increased aerosol load such as caused by, e.g., volcanic eruptions, wild fires, or dust storms. The highly sensitive altitude-dependent CI-thresholds will certainly also detect enhanced aerosol levels. Using additional detection and classification methods (e.g. Griessbach et al., 2016) can provide a distinction between ice clouds and aerosols with a different spectral signature. The synthetic studies below disregard aerosols for simplicity's sake.*"

Later on, we added to the 3-D retrieval from GLORIA data: *We determine a spatially resolved extinction value that does not allow to distinguish between trace gas, ice cloud, or aerosol emission. In the given spectral range, emission from trace gases is negligible compared to that of ice clouds, and we know from closer inspection of spectra and other sources of*

*information that there was no strong aerosol load such as generated by an volcanic eruption or biomass burning.*

6. *L138: It's not clear here — would the FTS produce data sampled only at the coarse resolution? Is the spectral resolution sufficient to do a good enough job with the gas retrievals? I'm assuming (from L125) that the emissivity growth approximation lookup tables are generated using high resolution line-by-line calculations and spectrally integrated to the coarse resolution. Is that the case? Some more detailed explanation seems necessary.*

The FTS takes interferograms of a certain length, which are processed to spectra of a certain sampling and resolution (the latter also depends on employed apodization). To relate the measured value to actual emissions, this information (and some more) can be used to compute the instrument line shape. The measurement corresponds then to the convolution of the 'real' continuous spectrum with the instrument line shape.

The IRLS in the discussed form would very quickly take interferograms of just 4 mm length, which corresponds to a spectral sampling of 1.25 cm$^{-1}$. Whether this is sufficient for trace gas retrievals is beyond the scope of this study, but other studies performed during the pre-selection period of EE7 (Kerridge et al., PREMIER Consolidation of Requirements and Synergistic Algorithms (CORSA) Study, Final Report, ESA, 2012) have examined this in great detail and came to satisfactory results for temperature, ozone, water vapour and some CFCs. From personal experience working with GLORIA data at 0.625 cm$^{-1}$, we see also no problems for processing actual data (compared to a synthetic study). We actually further degrade the spectral resolution of GLORIA by combining neighbouring samples for a faster retrieval, at the cost of some information and a slightly increased susceptibility to systematic errors. The retrieval approach for such a coarse spectral resolution is different compared to that employed for highly resolving instruments such as MIPAS and closer to, e.g., CRISTA or HIRDLS. The paper does not deal with trace gas retrievals and we do not think that this information will help here.

The tables for JURASSIC2 are indeed computed from line-by-line computations, the Oxford RFM model, to be precise. Those calculations were performed on a very fine grid and convolved with the (rather wide) instrument line shape functions.

We added *,,The tables were computed using a line-by-line model (Dudhia, 2017) convolved with the respective instrument line shapes, which for FTS spectrometers depend mostly on the interferogram length and employed apodization."*

7. *L145-L153: Please provide some information on the approximate optical depths corresponding to the CI values 1.1 and 4 and therefore what you mean by thin and thick clouds.*

Unfortunately there is no clear correlation between CI and optical depth in general, as the CI not only depends on the cloud extinction but also strongly on the background temperature and to some degree on trace gas concentrations. These impacts are reflected in the altitude and seasonally variable CI-thresholds for cloud detection. In our very limited CALIOP IRLS simulations, a CI value of 4.0 might correspond to a transmissivity of 0.86 or (optical depth of 0.14) in the atmospheric window at 833 cm$^{-1}$ and a CI value of 1.1 might correspond roughly to a transmissivity of 0.2 (optical depth 1.6) in the atmospheric window.

We added *,,Typical CI values in cloudy conditions are between 1.1 and 6 (Sembhi et al., 2012; Spang et al., 2012, 2015). The lowest detectable extinction from a space based limb emission measurement with the CI in the atmospheric window at 833 cm$^{-1}$) is of the order of 2×10$^{-5}$ km$^{-1}$ (Sembhi et al., 2012). For IR limb emission measurements the clouds are termed optically thick when the CI profile at and below cloud altitude runs into saturation, which occurs for extinctions of 5×10$^{-2}$ km$^{-1}$ and higher (Griessbach et al., 2016)."*

8. *L172: Maybe explain the concept of a "convex hull"*

We added ,,*The convex hull of a set of points is the smallest convex shape that fully encloses the points.*"

9. *L174: Maybe clarify this a bit better. What I am getting from this is that "below the threshold" means that the CI detection method got triggered. However, the CI detects a spurious cloud at the tangent, but it was triggered by a real cloud well above the tangent.*

This is correct. The CI method detects a cloud along the ray, but cannot say at which point along the ray. As such it is more capable of detecting cloud free spectra for trace gas processing than identifying positions of clouds. As crude approximation one can assume that the cloud is located at the tangent point, which is often, but not always correct.

We have expanded the introduction to the convex hull method: *If the cloud index of a measurement is below the threshold, a cloud has been detected along the line-of-sight of the measurement. But it is not clear where along the measurement the cloud is located. It could be, in theory, in low earth orbit, right in front of the satellite, close to the tangent point, or at any point in between or beyond. As such, the cloud index is better suited for selecting cloud-free measurements for trace gas retrievals than locating clouds. Due to the curvature of the earth, the line-of-sight of a measurement stays longer in the atmospheric layer surrounding the tangent point than those above. This makes it more sensitive to clouds in this layer than the layers above; but the signal of a 'thicker' cloud at a higher layer is still indistinguishable from a signal of a 'thinner' cloud in the tangent layer.*

10. *L181: "A common approximation of the spatial origin of a radiance measurement is assigning it to its tangent point, which is the location along its line-of-sight that is closest to the surface." The radiance measured is the result of the integration along the refracted ray path (which may also involve scattering of radiation into the line-of-sight). The tangent height is just a convenient label to use.*

Indeed. This is meant by "approximation". At least under optically thin conditions, the majority of radiation often stems from the tangent layer, typically shifted to some degree towards the instrument. In the UTLS, the atmosphere is often not perfectly thin anymore such that more and more radiation stems from the side of the ray path that is closer to the instrument.

However, the tangent point is commonly used as approximate origin. Obviously, it may not be the actual origin and if to call it an approximation or a label doesn't matter much in practice. We think it is important to point out that it is a potentially erroneous, but convenient, attribution.

The major advantage, in our eyes, of the extinction retrieval (and to a lesser extent the convex hull) is that it does away with the overly simplifying approximations such as the (in this case necessary) assignment of the radiance to the tangent point and allows inclusion of further instrument properties such as point spread function, etc.

11. *L197-L210: In this section (and everywhere else) it's not clear whether unrefracted or refracted tangent rays are being described. Refraction will cause the actual tangent point to be lowered and move closer to the instrument compared to the geometric (unrefracted) tangent point.*

Tangent points are always computed by JURASSIC2 using refraction. This is straightforward here due to the use of simulated data. Without refraction, significant errors would occur, especially below 15 km. The influence of temperature structure and pressure on refraction is comparatively small, though, such that simple correction methods can be applied to reduce the error to 10's of metres.

We added: ,,*JURASSIC2 is also used to compute lines-of-sight and tangent points in this study using a simple refraction scheme by Hase and Höpfner (1999).*"

12. *L216: The first reference to the CI being a "color-ratio" probably should be on L143.*

We moved the first reference upwards as suggested.

13. *L237: Eqn 1: The role of $\lambda_i$ is not defined. $F'$ is not mentioned.*

We amended the section to include the missing information.

14. *L250: 1/2 orbit in 25 mins or 1 day (typically 14.5 orbits) in 6 hrs.*

We added: *,,A full day of measurements (for an exemplary 14.5 orbits) would thus consume $\approx 6$ hours."*

15. *L284: What is the optical depth cut-off value?*

A cut off threshold is difficult to define, as it is a gradual descent into full opacity. Please see answer to comment 7. Assuming 1 km thick homogeneous ice clouds the CI profiles start to run into saturation for extinctions of $5 \times 10^{-2}\,\mathrm{km}^{-1}$ depending on cloud altitude, background atmosphere and cloud particle size. For extinctions of $1 \times 10^{-1}\,\mathrm{km}^{-1}$ all CI profiles saturate irrespective of the other properties.

Extinction values larger than $0.02\,\mathrm{km}^{-1}$ cause a transmissivity of $\approx 10\%$ after only 100 km. This is probably an extinction value that can be retrieved for clouds shorter than that. Values below $0.001\,\mathrm{km}^{-1}$ remain 75% transparent up to 300 km (typical length of a ray in a tangent layer) and can be derived without problems. We are in the process of examining the detection capabilities of such cloud retrievals and the usefulness of the derived extinction to determine, e.g., ice water path.

16. *L361: What is the clear-sky optical depth for the lowest tangent height you are considering?*

Actual tangent points values go down to 8 km due to refraction and some LOS-correction in the retrieval. Clear-sky optical path values go from $\approx 0.18$ at 8 km over $\approx 0.1$ at 9 km to 0.07 at 10 km.

17. *L392: I don't see how the retrieval could distinguish between spatial vs temporal evolution.*

Indeed, this is a problem of the tomography carried out on air planes that is greatly reduced by using a satellite with its much higher speed over ground.

In this particular case, Occam's razor tells us that a cloud close to the aircraft was responsible for the high radiance at all vertical angles (and the viscam for that matter), as the only alternative would be a humongous cloud forming spontaneously further away and disappearing quickly. The retrieval used an unchanging atmosphere as "a priori" knowledge and thus excludes this possibility for right or wrong. But even when assuming temporal change, this explanation is highly unlikely, and as such is discarded similar to many other potential solutions fitting to the measurements, but being physically implausible.

More rigorously, one can compute a "temporal" averaging kernel matrix to deduce the temporal resolution of deduced quantities similar to the spatial resolution (Ungermann et al., 2011). The temporal resolution here is in the order of dozens of minutes with obvious problems. Using this, one can assign a temporal interval to points of the 3-D volume. Given the temporal resolution of available models and observations, this was however not terribly useful, also taking into account that computing the 4-D averaging kernel matrix is only feasible for selected atmospheric samples for practical reasons.

18. *L426: What about evaporation?*

Indeed. Evaporation and condensation are also processes that complicate the comparison. We extended together *,,In addition, we used the cloud top product from the MODIS instrument that passed over our measurement area in low Earth orbit around 13:08 UTC, which is*

*only slightly beyond GLORIA's measurement period, and still reasonably close to compare the cloud top altitudes. However, the horizontal position might have been shifted due to advection and small clouds may also appear and disappear due to condensation and evaporation."*

[revised manuscript text omitted]